# QuadAttac$K$: A Quadratic Programming Approach to Learning Ordered Top-$K$ Adversarial Attacks

**Thomas Paniagua, Ryan Grainger and Tianfu Wu**
Department of Electrical and Computer Engineering, NC State
{tapaniag, rpgraing, twu19}@ncsu.edu
https://thomaspaniagua.github.io/quadattack_web/

## Abstract

The adversarial vulnerability of Deep Neural Networks (DNNs) has been well-known and widely concerned, often under the context of learning top-1 attacks (e.g., fooling a DNN to classify a cat image as dog). This paper shows that the concern is much more serious by learning significantly more aggressive ordered top-$K$ clear-box [1] targeted attacks proposed in [Zhang and Wu, 2020]. We propose a novel and rigorous quadratic programming (QP) method of learning ordered top-$K$ attacks with low computing cost, dubbed as **QuadAttac$K$**. Our QuadAttac$K$ directly solves the QP to satisfy the attack constraint in the feature embedding space (i.e., the input space to the final linear classifier), which thus exploits the semantics of the feature embedding space (i.e., the principle of class coherence). With the optimized feature embedding vector perturbation, it then computes the adversarial perturbation in the data space via the vanilla one-step back-propagation. In experiments, the proposed QuadAttac$K$ is tested in the ImageNet-1k classification using ResNet-50, DenseNet-121, and Vision Transformers (ViT-B and DEiT-S). It successfully pushes the boundary of successful ordered top-$K$ attacks from $K = 10$ up to $K = 20$ at a cheap budget ($1 \times 60$) and further improves attack success rates for $K = 5$ for all tested models, while retaining the performance for $K = 1$.

## 1 Introduction

As the development of machine deep learning and Artificial Intelligence (AI) continues to accelerate, the need to address potential vulnerabilities of Deep Neural Networks (DNNs) becomes increasingly crucial towards building safe-enabled and trustworthy learning and AI systems. Among these vulnerabilities, adversarial attacks [Xie et al., 2017, Kos et al., 2018, Sharif et al., 2016, Ebrahimi et al., 2018, Qin et al., 2019, Lin et al., 2017, Papernot et al., 2017, Liu et al., 2017, Xie et al., 2019b, Dong et al., 2019, Goodfellow et al., 2015, Kannan et al., 2018, Madry et al., 2017, Xie et al., 2019a], that can almost arbitrarily manipulate the prediction of a trained DNN for a given testing data by learning visually imperceptible perturbations, especially under clear-box target attack settings, are of particular interest in terms of revealing the shortcut learning of discriminatively-trained DNNs [Geirhos et al., 2020]. Clear-box targeted adversarial attacks are often learned under the top-1 attack setting with the objective of causing catastrophic performance drop of the top-1 accuracy on adversarial examples against the clean counterparts.

To generalize the vanilla setting of learning top-1 attacks, much more aggressive **ordered top-$K$ attacks** have been proposed in [Zhang and Wu, 2020], which aim to learn adversarial examples that *manipulate a trained DNN to predict any specified $K$ attack targets in any given order as the top-$K$ predicted classes*. As shown in Fig. 5, in the ImageNet-1k [Russakovsky et al., 2015] classification

---

[1]This is often referred to as white/black-box attacks in the literature. We choose to adopt neutral terminology, clear/opaque-box attacks in this paper, and omit the prefix clear-box for simplicity.

37th Conference on Neural Information Processing Systems (NeurIPS 2023).

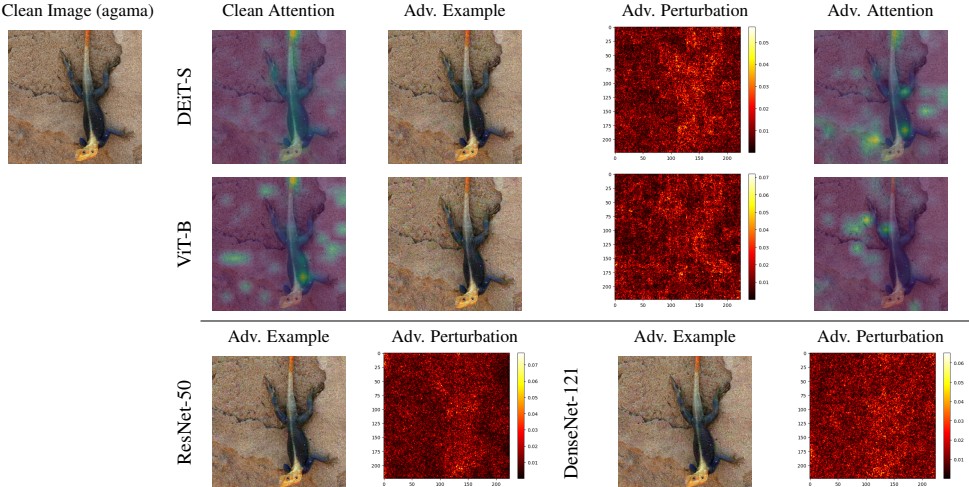

Figure 1: Adversarial examples learned by our QuadAttac$K$ ($K$ = 20) for a same clean image using four different networks: two popular convolutional neural networks (ResNet-50 [He et al., 2016] and DenseNet-121 [Huang et al., 2017]), and two widely used Vision Transformers (ViT-B [Dosovitskiy et al., 2020] and DEiT-S [Touvron et al., 2021]). The ground-truth label is *agama*. **The ordered top-20 targets** (randomly sampled and kept unchanged for the four models) are: [sea cucumber, barrow, odometer, bloodhound, hen-of-the-woods, ringneck snake, snail, tiger shark, Pembroke, altar, wig, submarine, macaw, combination lock, ram, Irish wolfhound, confectionery, buckle, chime, garden spider]. For example, the top-20 classes of the clean image by DEiT-S are [*agama, frilled lizard, shovel, reel, common iguana, Yorkshire terrier, coho, alligator lizard, hand blower, meerkat, ostrich, mongoose, fiddler crab, eft, wing, bustard, green lizard, whiptail, brass, American chameleon*], which have more or less visual similarities. More examples are the Appendix B.

task, consider a clean testing image of 'Agama' that can be correctly classified by some trained DNNs such as the four networks we tested, let's say a desired list of the ordered top-20 attack targets as shown in the caption. An ordered top-$K$ attack method would seek to find the adversarial perturbation to the input image that manipulates the DNN's predictions to ensure that the top-20 predicted classes of the perturbed image match the specified list in the given order.

**Why to learn ordered top-$K$ attacks?** They facilitate exploiting the principle of "class coherence" that reflect real-world scenarios where the relative importance or priority of certain classes is crucial to recognize the relationships or logic connecting classes within ordinal or nominal context. Unlike unordered top-$K$ or top-1 attacks, ordered top-$K$ attacks can subtly manipulate predictions while maintaining coherence within the expected context.

- **An Ordinal Example.** Imagine a cancer risk assessment tool that analyzes 2D medical images (e.g., mammograms) to categorize patients' cancer risk into the ordinal 7-level risk ratings (*[Extremely High Risk, Very High Risk, High Risk, Moderate Risk, Low Risk, Minimal Risk, No Risk]*), An oncologist could use this tool to triage patients, prioritizing those in the highest risk categories for immediate intervention. An attacker aiming to delay treatment might use *an ordered top-3 adversarial attack* to change a prediction for a patient initially assessed as Very High Risk. They could target the classes *[Moderate, Low, Minimal]*, subtly downgrading the urgency without breaking the logical sequence of risk categories. An unordered attack, in contrast, might lead to a sequence like *[Low, Very High, Minimal]*, disrupting the ordinal relationship between classes. Such a disruption could raise red flags, making the attack easier to detect.

- **A Nominal Example.** Traffic control systems could use deep learning to optimize flow by adjusting the timing of traffic lights based on the types of vehicles seen. Priority might be given to certain vehicle classes, such as public transit or emergency vehicles, to improve response times. Imagine a city's traffic control system, which has specific traffic light timing behavior for the nominal vehicle categories *[Emergency Vehicle, Public Transit, Commercial Vehicle, Personal Car, Bicycle]*. Public transit might be given slightly extended green lights during rush hours to encourage public transportation use. An attacker wanting to cause delays for personal cars without raising alarms could launch *an ordered top-2 adversarial attack*, targeting the sequence *[Commercial Vehicle, Public Transit]*. This would cause the system to interpret most personal cars as commercial vehicles during the attack, applying the extended green light times meant for public transit to lanes primarily

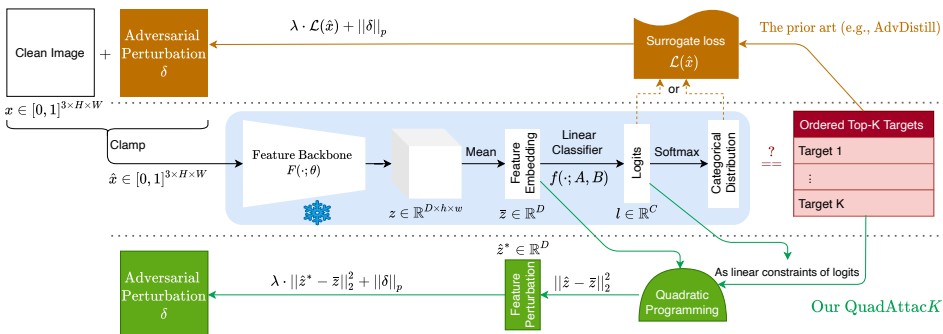

Figure 2: Illustration of the proposed QuadAttack$K$ method in comparison with the prior art (e.g., the adversarial distillation (AD) method [Zhang and Wu, 2020]).

used by commercial vehicles. An unordered top-2 attack that may result in *[Emergency Vehicle, Commercial Vehicle]*, would likely be quickly detected, as emergency vehicle priority changes are significant and could be easily noticed by traffic operators (this weakness is exacerbated in any top-1 attack or unordered attacks).

**Successful ordered top-$K$ attacks can potentially provide several advantages:** enabling better controllability in learning attacks that are more difficult to detect, revealing deeper vulnerability of trained DNNs, and testing the robustness of an attack method itself, especially when $K$ is relatively large (e.g., $K \geq 15$) and the computing budget is relatively low (e.g., 60 steps of optimization).

**Learning ordered top-$K$ attacks is an extremely challenging problem.** The adversarial distillation method [Zhang and Wu, 2020] is the state of the art method (see Sec. 3.1), which presents a heuristic knowledge-oriented design of the ordered top-$K$ target distribution, and then minimizes the Kullback-Leibler divergence between the designed distribution and the DNN output distribution (after softmax). It often completely fails when $K > 10$. In this paper, we show it is possible to learn those attacks for a variety of DNNs, including ResNet-50 [He et al., 2016], DenseNet-121 [Huang et al., 2017] and Vision Transformers [Dosovitskiy et al., 2020, Touvron et al., 2021].

**The key to learning clear-box targeted attacks ($K \geq 1$) lies in the objective function for optimization**, which usually consists of two terms, one is the $\ell_p$ norm (e.g., $\ell_2$) of the learned adversarial perturbation (to be as small as possible to be visually imperceptible), and the other is the surrogate loss capturing the specified attack constraints such as the top-$K$ extended C&W (hinge) loss [Carlini and Wagner, 2017] and the adversarial distillation loss proposed in [Zhang and Wu, 2020] (see Sec. 3.1). The trade-off between the two terms are often searched in optimization with respect to a certain computing budget (e.g., $9 \times 30$ means to test 9 different trade-off parameter assignments based on linear search in a predefined range, and to run 30 forward-backward computation iterations of the DNN per trade-off parameter search step). After the optimization, Attack Success Rates (ASR, higher is better) and some $\ell_p$ norms (e.g., $\ell_1$ and $\ell_2$) of learned successful perturbations (smaller is better) are used as evaluation metrics. As illustrated in Fig. 2, in this paper, we propose a novel formulation which is different from the prior art in the three aspects as follows:

- We identify that while sufficient to capture the top-$K$ attack constraint, hand-crafted surrogate losses are not necessary and often introduce inconsistency and artifacts in optimization (see Sec. 3.2). We eliminate the need of introducing surrogate losses. Instead, we keep the top-$K$ attack constraints in the vanilla form and cast the optimization problem as quadratic programming (QP). We solve the QP by leveraging a recently proposed differentiable QP layer (for PyTorch) [Amos and Kolter, 2017]. We present an efficient implement to parallelize the batched QP layer for any ordered top-$K$ targets specified individually for each instance in a batch.
- We observe that directly minimizing the $\ell_p$ norm of learned perturbations together with the hand-crafted surrogate loss could miss the chance of exploiting semantic structures of the feature embedding space (i.e., the input to the final linear classifier). Instead, we minimize the Euclidean distance between the feature embedding vectors at two consecutive iterations in the optimization. *This can be understood as the latent perturbation learning versus the raw data perturbation learning*(see Sec. 3.3). Our proposed latent perturbation learning enables more consistent optimization trajectories in pursuing the satisfaction of the specified top-$K$ attack constraints. The minimized Euclidean distance is then used as the loss together with the $\ell_p$ norm of the learned perturbation in computing the adversarial perturbation via back-propagation at each iteration.

## 2  Related Work and Our Contributions

**Adversarial Attacks** have remained as a critical concern for DNNs, where imperceptible perturbations to input data can lead to significant misclassification [Xie et al., 2017, Hendrik Metzen et al., 2017, Chen et al., 2019, Liu et al., 2018]. Various approaches have been proposed to investigate the vulnerabilities of DNNs and exploit their sensitivity to non-robust features. Notable works include the seminal discovery of visually imperceptible adversarial attacks [Szegedy et al., 2013], which highlighted the need to evaluate the brittleness of DNNs and address it with explicit defense methods [Madry et al., 2017, Cohen et al., 2019, Wang et al., 2020, Wong et al., 2020]. Clear-box attacks, which assume full access to the DNN model, have been particularly effective in uncovering vulnerabilities [Madry et al., 2017]. The C&W method [Carlini and Wagner, 2017], for instance, introduced loss functions that have been widely adopted to generate adversarial examples. Momentum-based methods like [Dong et al., 2018, Lin et al., 2019] and gradient projection techniques like PGD [Madry et al., 2017] have also been successful in crafting adversarial examples.

**Unordered top-$K$ Attacks** aim to manipulate the top-$K$ predicted classes of a DNN without enforcing a specific order among them. A common approach is to optimize a loss function that combines multiple objectives, such as maximizing the probability of the target classes while minimizing the probabilities of other classes simultaneously. [Tursynbek et al., 2022] presents a geometry inspired method that allows taking gradient steps in favor of simultaneously maximizing all target classes while maintaining a balance between them. Other approaches, for example, may use sorting strategies [Kumano et al., 2022] to limit the set of logits involved simultaneously in a loss and target specific logits that contribute most to a misclassification. The "Superclass" attacks have been proposed in [Kumano et al., 2022], for which ordered top-$K$ attacks can been seen as a generalization. Non-targeted top-$K$ attacks are studied in [Zhang et al., 2022].

**Ordered top-$K$ Attacks** [Zhang and Wu, 2020] require preserving both the attack success rate and the order of the top-$K$ predicted classes. Following the intuitions from [Kumano et al., 2022] we note that adversarial sample detection and defense methods [Lee et al., 2018, Huang and Li, 2021, Aldahdooh et al., 2022] may benefit from the fact that many adversarial attacks tend to generate a nonsensical class prediction (e.g. the top-$K$ predictions may all be from a different super-class [Kumano et al., 2022]). An ordered top-$K$ adversarial attack may choose to create an attack that is semantically plausible and have a higher potential to fool detection methods and defense methods. The sorting constraint in ordered top-$K$ attacks adds a layer of complexity to the optimization problem. By enforcing a specific order among the top-$K$ predictions, the attacker must not only manipulate the logits to maximize the target classes' probabilities but also ensure that the predicted order aligns with the desired order. In [Zhang and Wu, 2020], two methods are presented through the use of semantic information in the form of class language embedding.

**Constrained Optimization in DNNs**, such as OptNet [Amos and Kolter, 2017] and other differentiable optimization works [Agrawal et al., 2019, Butler and Kwon, 2023], have introduced powerful formulations for integrating optimization layers within the network. These types of works have produced many DNN based problem solutions while also being able to use domain knowledge to constrain solutions and reduce data requirements [Sangalli et al., 2021] and solve problems otherwise intractable with DNNs [Wang et al., 2019, Mandi et al., 2020]. While OptNet's quadratic solver is typically used as a layer, our focus is on attacking a pre-trained model with a fixed architecture. Thus, we leverage the principles of constrained optimization to formulate an objective function that captures the constraints of ordered top-$K$ adversarial attacks. This adaptation allows us to guide the attack process to satisfy the ordering constraints while maximizing target class probabilities, which provides new insights for enhancing attack effectiveness and robustness.

**Our Contributions** This paper makes two main contributions to the field of learning clear-box targeted adversarial attacks:

- It presents a quadratic programming (QP) approach to learning ordered top-$K$ attacks, dubbed as QuadAttac$K$. It eliminates hand-crafting surrogate loss functions to capture the top-$K$ attack constraint. It provides a QP based formulation to better exploit the semantics of the feature embedding space in capturing the top-$K$ attack requirement.
- It obtains state-of-the-art adversarial attack performance in ImageNet-1k classification using both ConvNets (ResNet-50 and DenseNet-121) and Transformer models (ViT-B and DEiT-S). It pushes the limit of the number of targets, $K$ to a large number, for which the prior art completely fails.

# 3 Approach

In this section, we first define the problem of learning ordered top-$K$ attacks following [Zhang and Wu, 2020], as well as the extended top-$K$ C&W (CW$^K$) method and the adversarial distillation (AD) method. We then present details of our proposed QP based formulation, i.e., QuadAttac$K$.

## 3.1 Learning Ordered top-$K$ Clear-Box Targeted Attacks: the Problem and the Prior Art

We consider image classification DNNs, which consist of a feature backbone and a head linear classifier. Denote by $F : x \in \mathbb{R}^{3 \times H \times W} \to z \in \mathbb{R}^{D \times h \times w}$ the feature backbone which transforms an input image to its feature map, where an input image $x$ is often a RGB image of the spatial height and width, $H$ and $W$ (e.g., $224 \times 224$), clean or perturbed, and the feature map $z$ is in a $D$-dim feature space with the the spatial height and width, $h$ and $w$ (e.g., $7 \times 7$) based on the overall spatial downsampling stride implemented in the feature backbone. Let $\bar{z} \in \mathbb{R}^D$ be feature embedding vector for $x$ via any spatial reduction method (e.g. global average pooling). Denote by $f : \bar{z} \in \mathbb{R}^D \to l \in \mathbb{R}^C$ the linear head classifier, where $C$ is the number of classes (e.g., 1000 in ImageNet-1k [Russakovsky et al., 2015]), and $l$ is the output logit vector. We have,

$$l = f(F(x; \theta); A, B) = A \cdot \bar{z} + B, \tag{1}$$

where $\theta$ collects all the model parameters of the feature backbone, and $A \in \mathbb{R}^{C \times D}$ and $B \in \mathbb{R}^C$ the weight and bias parameters of the head linear classifier. In learning clear-box attacks, we assume all the information of the network are available, and the parameters $(\theta, A, B)$ are frozen throughout.

Denote by $(x, y)$ a pair of clean image and its ground-truth label ($y \in \mathcal{Y} = \{1, 2, \cdots, C\}$). For learning attacks, we assume $x$ can be correctly classified by the network, i.e., $y = \arg\max_i l_i$. Denote by $T$ the ordered list of attack target(s) with the cardinality $K = |T|$, where the ground-truth label is excluded, $y \notin T$, and by $T^c = \mathcal{Y} \setminus T$ the complement set. Let $\delta(x, T; F, f)$ be the adversarial perturbation to be learned. **An ordered top-$K$ adversarial example** is defined by $\hat{x} = x + \delta(x, T; F, f)$ if the top-$K$ prediction classes for $\hat{x}$ equal to $T$ based on its logits $\hat{l}$ ( Eqn. 1).

Learning ordered top-$K$ attacks is often posed as a constrained optimization problem,

$$\underset{\delta}{\text{minimize}} \quad ||\delta||_p, \tag{2}$$

$$\text{subject to} \quad \hat{l}_{t_i} > \hat{l}_{t_{i+1}}, \quad i \in [1, K-1], \quad t_i \in T,$$
$$\hat{l}_{t_K} > \hat{l}_j, \qquad t_K \in T, \quad \forall j \in T^c,$$
$$\hat{x} = x + \delta \in [0, 1]^{3 \times H \times W},$$
$$\hat{l} = f(F(\hat{x}; \theta); A, B),$$

where the first two constraints capturing the ordered top-$K$ attack requirement. This traditional formulation leads to the challenge in optimization, even with $K = 1$ as pointed out in the vanilla C&W method [Carlini and Wagner, 2017]. Some surrogate loss, $\mathcal{L}(\hat{x})$, is necessary to ensure the first two terms are satisfied when $\mathcal{L}(\hat{x})$ is minimized. We have,

$$\text{minimize} \quad \lambda \cdot \mathcal{L}(\hat{x}) + ||\delta||_p, \tag{3}$$

$$\text{subject to} \quad \hat{x} = x + \delta \in [0, 1]^{3 \times H \times W},$$

where $\lambda$ is the trade-off parameter between the visual imperceptibility of learned perturbations and the ASR. The remaining constraint can be addressed via a projected descent in optimization. So, the optimization can enjoy the straightforward back-propagation algorithm that is used in training the DNN on clean images.

The extended top-$K$ C&W (hinge) loss [Zhang and Wu, 2020] is defined by,

$$\mathcal{L}_{CW}^K(\hat{x}) = \sum_{i=1}^{K} \max\left(0, \max_{j \in \mathcal{Y} \setminus \{t_1, \cdots, t_i\}} \hat{l}_j - \min_{t \in \{t_1, \cdots, t_i\}} \hat{l}_t\right). \tag{4}$$

And, the adversarial distillation loss [Zhang and Wu, 2020] is defined by,

$$\mathcal{L}_{AD}^K(\hat{x}) = KL(\hat{p}||P^{AD}) = \sum_{t_i \in T} \hat{p}_{t_i}(\log \hat{p}_{t_i} - \log P_{t_i}^{AD}) + \sum_{j \in T^c} \hat{p}_j(\log \hat{p}_j - \log P_j^{AD}), \tag{5}$$

where $\hat{p} = \text{Softmax}(\hat{l})$ and $P^{AD}$ is the knowledge-oriented heuristically-designed adversarial distribution with the top-$K$ constraints satisfied (see [Zhang and Wu, 2020] for details). $KL(\cdot||\cdot)$ is the Kullback-Leibler divergence between two distributions.

## 3.2 Limitations of the Prior Art

From the optimization perspective, we observe there are two main drawbacks in the aforementioned formulations (Eqns. 3, 4, 5):

- The two surrogate loss formulations (Eqns. 4 and 5) are sufficient, but not necessary. They actually introduce inconsistency and artifacts in optimization. The extended top-$K$ C&W loss in Eqn. 4 is not aware of, and thus can not preserve, the subset of targets whose relative order has been satisfied. For example, consider there are 5 classes in total, and a specified ordered top-3 list of targets, $[2, 3, 1]$. Assume at a certain iteration, the predicted classes for $\hat{x}$ in sort are $[4, 2, 3, 5, 1]$, in which the relative order of the specified 3 targets has been satisfied. The loss $\mathcal{L}^3_{CW}(\hat{x}) = \sum_{i=1}^3 (\hat{l}_4 - \hat{l}_i)$, which mainly focuses on pushing down the logit $\hat{l}_4$ and/or pulling up the logits, $\hat{l}_i$'s ($i = 1, 2, 3$). So, at the next iteration, it may results in the sorted prediction like $[1, 3, 2, 4, 5]$, leading to a totally wrong relative order. The adversarial distillation loss in Eqn. 5 has similar problems, i.e., the first part, $\sum_{i \in T} \hat{p}_i (\log \hat{p}_i - \log P_i^{AD})$ is not aware of some satisfied relative order. It further enforces order between non-target classes since the adversarial distribution $P^{AD}$ needs to be specified before the optimization, as shown in the second part, $\sum_{j \in T^c} \hat{p}_j (\log \hat{p}_j - \log P_j^{AD})$.

- Directly minimizing $||\delta||_p$ in Eqn. 3 (i.e., adversarial learning in the data space) may actually hinder the effectiveness of learning adversarial examples due to the fact that the $\ell_p$ norm is totally unaware of the underlying data structures in the complex data space. Since a trained DNN is kept frozen in learning attacks, we can first perform adversarial learning in the feature embedding space (i.e., the head linear classifier's input space, $\bar{z}$ in Eqn. 1), which has been learned to be discriminatively and/or semantically meaningful. With a learned adversarial perturbation for $\bar{z}$, we can easily compute the perturbation for the input data.

We address these limitations in this paper by proposing a novel QP based formulation – QuadAttack$K$. We present the detail of our QuadAttac$K$ in the following sub-sections.

## 3.3 The Proposed QuadAttac$K$

We first show that any specified ordered top-$K$ attack requirements, $T$, can be cast as linear constraints in a compact matrix form, denoted by $D_T$. Consider $\ell_2$ norm of Eqn. 2, we can rewrite it as,

$$\underset{\delta}{\text{minimize}} \quad ||\hat{x} - x||_2^2, \tag{6}$$

$$\text{subject to} \quad D_T \cdot \hat{l} > 0, \quad D_T \in \{-1, 0, 1\}^{C-1 \times C},$$
$$\hat{x} = x + \delta \in [0, 1]^{3 \times H \times W},$$
$$\hat{l} = f(F(\hat{x}; \theta); A, B),$$

where $D_T$ is a $C - 1 \times C$ matrix constructed from the specified targets $T$, and $C$ is the number of classes in total. Consider the aforementioned toy example where $C = 5$ and the ordered top-3 attack targets are $T = [2, 3, 1]$, we have,

$$
\begin{aligned}
\hat{l}_2 - \hat{l}_3 &> 0, \\
\hat{l}_3 - \hat{l}_1 &> 0, \\
\hat{l}_1 - \hat{l}_4 &> 0, \\
\hat{l}_1 - \hat{l}_5 &> 0,
\end{aligned}
\tag{7}
\qquad
D_T \cdot \hat{l} > 0, \quad D_T =
\begin{bmatrix}
0 & 1 & -1 & 0 & 0 \\
-1 & 0 & 1 & 0 & 0 \\
1 & 0 & 0 & -1 & 0 \\
1 & 0 & 0 & 0 & -1
\end{bmatrix},
\tag{8}
$$

where Eqn. 7 is the vanilla form of expressing the specified top-3 attack requirements, and Eqn. 8 is the equivalent compact matrix form.

However, Eqn. 6 can not be easily solved via QP due to the highly nonconvex nature of the feature backbone $F$ in the third term of the constraints. This nonconvexity hinders the ability to solve the problem and find an optimal solution while satisfying constraints. Eqn. 6 aims to directly seek the adversarial perturbations in the data space, which has to include the nonconvex feature backbone $F$ in the optimization. Since $F$ is a frozen transformation in learning attacks, and the head classifier $f$ is a linear function, **we can separate the learning of ordered top-$K$ attacks in two steps:**

i) We first satisfy the ordered top-$K$ constraints without resorting to hand-crafted surrogate losses via QP in the feature embedding space (i.e., the output space of $F$ and the input space of $f$). The last

constraint in Eqn. 6 will be replaced by $\hat{l} = A \cdot \bar{z} + B$ (see Eqn. 1). Denote by $\delta$ the image perturbation at the current iteration. We have the current feature embedding vector $\bar{z} = \text{Mean}(F(x + \delta; \theta))$. our proposed QuadAttac$K$ first solves the perturbed $\hat{z}$ via QP,

$$\underset{\hat{z}}{\text{minimize}} \quad ||\hat{z} - \bar{z}||_2^2, \tag{9}$$

$$\text{subject to} \quad D_T \cdot \hat{l} > 0,$$

$$\hat{l} = A \cdot \hat{z} + B.$$

ii) After having found $\hat{z}$ we then use the residual (Euclidean distance) $||\bar{z} - \hat{z}||_2^2$ as the loss to find the perturbed image $\hat{x}$. Specifically, with the optimized $\hat{z}$ (i.e., the closest point in latent space where top-$K$ attack constraints are satisfied), we compute the image perturbation using vanilla one-step back-propagation with a learning rate $\gamma$ and a loss weighting parameter $\lambda$. We find that both $\gamma$ or $\lambda$ may be used to control the tradeoff between attack success and magnitude of the perturbation found (see Fig. 3 and Appendix A). We have,

$$\delta^* = \delta - \gamma \cdot \frac{\partial}{\partial \delta} \left( \lambda \cdot ||\hat{\bar{z}} - \bar{z}||_2 + ||\delta||_p \right), \tag{10}$$

$$\hat{\delta} = \text{Clamp}(x + \delta^*; 0, 1) - x, \quad \hat{x} = x + \hat{\delta},$$

where $\text{Clamp}(\cdot; 0, 1)$ is an element-wise projection of the input onto $[0, 1]$.

**Understanding QuadAttac$K$:** In our QuadAttac$K$, the matrix $A$ plays a crucial role in encoding useful semantic class relationships. For instance, if classes like "cat" and "building" have distinct and separate representations in the latent space, the matrix $A$ will reflect these differences. As a result, the optimization problem would naturally prioritize target logits that have high activations for either "cat" or "building," but not both simultaneously. This semantic constraint in the feature embedding space helps guide the search towards relevant and meaningful perturbations that maintain the desired order constraints while avoiding conflicting activations for disparate classes. By incorporating the matrix $A$ to capture semantic relationships, our QuadAttac$K$ not only overcomes the nonconvexity challenge of the original problem but also leverages meaningful class information to guide the search and generate effective adversarial perturbations.

## 3.4 Fast and Parallel Quadratic Programming Solutions

An efficient solver is crucial for addressing the QP formulation of learning ordered top-$K$ adversarial attacks. To that end, a fast, parallel, and GPU-capable quadratic programming solver is required. In this context, the *qpth* package created by [Amos and Kolter, 2017] emerges as a suitable choice, providing a PyTorch-enabled differentiable quadratic programming solver that enables efficient optimization while harnessing the power of GPUs,

$$\underset{\hat{z}}{\text{minimize}} \quad \frac{1}{2}\hat{z}^\top Q \hat{z} + p^\top \hat{z} \tag{11}$$

$$\text{subject to} \quad G\hat{z} \leq h,$$

$$W\hat{z} = b.$$

The main challenge lies in transforming our current formulation into one compatible with the *qpth* package (Eqn. 11). This involves structuring the objective and constraints to match the required standard form, as well as building the required matrices $Q, G$ and $h$ from attack targets in an efficient and parallel way. Finding $Q$ and $p$ is straightforward from an expansion of our squared Euclidean distance objective since $||\bar{z} - \hat{z}||_2^2 = \hat{z}^\top \hat{z} - 2\bar{z}^\top \hat{z} + \bar{z}^\top \bar{z}$. The term $\bar{z}^\top \bar{z}$ is a constant in our optimization so we can just consider the optimization of $\hat{z}^T \hat{z} - 2\bar{z}^T \hat{z}$. From this we can trivially see $Q = 2I$ where $I$ is the identity matrix and $p = -2\bar{z}^\top$. Further, since we need no equality constraints $W = 0, b = 0$. To formulate $G$ and $h$ we can rewrite the constraints in Eqn 9 as follows,

$$D_T \cdot (A\hat{z} + B) > 0 \quad \Rightarrow \quad -D_T \cdot A\hat{z} \leq D_T \cdot B - \eta, \tag{12}$$

where $\eta$ is the slack variable which is a small non-zero constant to allow our constraints to define a closed-convex set allowing equality in our constraint but still maintaining top-$K$ order when the constraint is satisfied. We have found $\eta = 0.2$ is an acceptable value, but other small values will also work. Intuitively if our current value for $\hat{z}$ does not satisfy our top-$K$ constraints then $\eta = 0$ would find a $\hat{z}$ on the boundary of the latent space that satisfies our constraints. The higher $\eta$ is the further away from the boundary and inside the set $\hat{z}$ becomes. From the above rearrangement we can easily see $G = -D_T \cdot A$ and $h = D_T \cdot B - \eta$ in Eqn. 11.

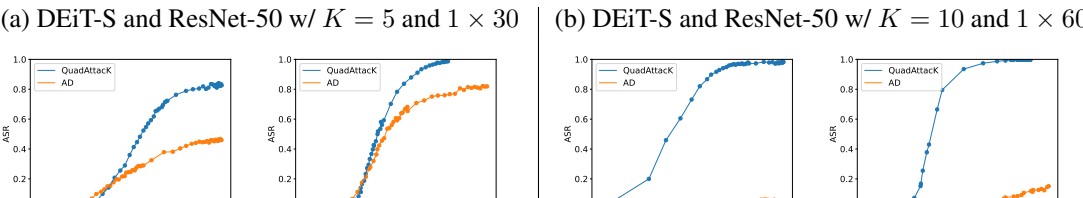

(a) DEiT-S and ResNet-50 w/ $K = 5$ and $1 \times 30$   (b) DEiT-S and ResNet-50 w/ $K = 10$ and $1 \times 60$

Figure 3: ASR vs $\ell_2$ energy tradeoff curves, which holistically compare the capacity of our QuadAttac$K$ against the prior art – the adversarial distillation method [Zhang and Wu, 2020], verifying our QuadAttac$K$'s advantages.

## 4 Experiments

In this section, we evaluate our QuadAttac$K$ with $K = 1, 5, 10, 15, 20$ in the ImageNet-1k benchmark [Russakovsky et al., 2015] using two representative pretrained ConvNets: the ResNet-50 [He et al., 2016] and the DenseNet-121 Huang et al. [2017], and two representative pretrained Transformers: the vanilla Vision Transformer (Base) [Dosovitskiy et al., 2020] and the data-efficient variant DEiT (small) [Touvron et al., 2021]. The ImageNet-1k pretrained checkpoints of the four networks are from the `mmpretrain` package [Contributors, 2023], in which we implement our QuadAttac$K$ and re-produce both CW$^K$ and AD [Zhang and Wu, 2020].

**Data and Metric.** In ImageNet-1k [Russakovsky et al., 2015], there are $50,000$ images for validation. To study attacks, we utilize the subset of images for which the predictions of all the four networks are correct. To reduce the computational demand, we randomly sample a smaller subset following [Zhang and Wu, 2020]: We iterate over all 1000 categories and randomly sample an image labeled with it, resulting in 1000 test images in total. For each $K$, we randomly sample 5 groups of $K$ targets with ground-truth (GT) label exclusive for each image in our selected test set, and then compute the `Best`, `Mean` and `Worst` ASRs, as well as the associated $\ell_1, \ell_2$ and $\ell_\infty$ energies. We mainly focus on low-cost learning of attacks (within 60 steps of optimization) for better practicality and better understanding of the underlying effectiveness of different attack methods. We also learn attacks with higher budgets ($9 \times 60$ and $9 \times 30$) for $K = 5, 10$. We use a default value for the trade-off parameter $\lambda$ in Eqn. 3 and Eqn. 10. *Details are provided in the Appendix A and our released code repository.*

**Baselines.** We compare our QuadAttac$K$ with previous state-of-the-art methods, namely, the top-$K$ extended CW$^K$ method and the Adversarial Distillation (AD) proposed in [Zhang and Wu, 2020]. We reproduce them in our code repository and test them on the four networks under the same settings for fair comparisons.

**An important detail in optimization:** In optimization, perturbations are initialized with some small energy white Gaussian noise. During the initial steps of optimization, the optimizer takes steps with large increases in perturbation energy since happens to be away from many required energies for a successful attack. These large increases in energy induces a momentum in the optimizer, which makes it difficult to reduce L2 energy in future iterations even if our objective function's gradient points towards a direction with minimal energy. By introducing a small number of warmup steps (e.g., 5, as commonly done in training a network on ImageNet from scratch) after which the optimizer's state is reset, we observe the performance of all analyzed methods are significantly improved. Our QuadAttac$K$ benefits most.

**Results.** The comparison results are shown in Table 1. Our proposed QuadAttac$K$ retains performance comparably for $K = 1$, and obtains consistently better performance, often by a large margin, for $K = 5, 10, 15, 20$. Especially, it addresses the challenges associated with large values of $K$ (e.g. $K \geq 10$) under the low-cost budget setting ($1 \times 60$). The prior art completely fails, while our QuadAttac$K$ can still obtain appealing ASRs.

**Analyses on the Trade-Off Between ASRs and Attack Energies.** Since we have two metrics (ASR and attack energy), the trade-off between them needs to be compared in order to comprehensively understand different methods. The trade-off curves (Fig. 3) explore the concept of how a higher success rate may be achieved by choosing to have higher energies and conversely a lower energy may be achieved by choosing to have a lower success rate. They holistically compare the capacity of QuadAttac$K$ against the prior art – the adversarial distillation method [Zhang and Wu, 2020].

| Protocol | Attack Method | Best ASR↑ | ℓ₁↓ | ℓ₂↓ | ℓ∞↓ | Mean ASR↑ | ℓ₁↓ | ℓ₂↓ | ℓ∞↓ | Worst ASR↑ | ℓ₁↓ | ℓ₂↓ | ℓ∞↓ |
|---|---|---|---|---|---|---|---|---|---|---|---|---|---|
| | | | | | | ResNet-50 [He et al., 2016] | | | | | | | |
| Top-20 | **QuadAttac**$K_{1\times60}$ | 0.2760 | 2350.79 | 7.71 | 0.0941 | 0.2546 | 2363.38 | 7.74 | 0.0939 | 0.2420 | 2381.09 | 7.80 | 0.0936 |
| Top-15 | **QuadAttac**$K_{1\times60}$ | 0.9520 | 1960.29 | 6.45 | 0.0763 | 0.9400 | 1957.23 | 6.44 | 0.0763 | 0.9290 | 1963.93 | 6.46 | 0.0767 |
| | **QuadAttac**$K_{1\times30}$ | 0.3640 | 2275.78 | 7.31 | 0.0711 | 0.3270 | 2279.91 | 7.32 | 0.0710 | 0.3050 | 2266.93 | 7.28 | 0.0706 |
| Top-10 | $AD_{1\times60}$ | 0.1150 | 1584.19 | 5.23 | 0.0687 | 0.1002 | 1576.03 | 5.21 | 0.0685 | 0.0880 | 1578.91 | 5.22 | 0.0682 |
| | **QuadAttac**$K_{1\times60}$ | **0.9990** | **1538.45** | **5.08** | **0.0560** | **0.9982** | **1534.04** | **5.06** | **0.0561** | **0.9970** | **1527.30** | **5.04** | **0.0559** |
| | **QuadAttac**$K_{1\times30}$ | 0.9640 | 1789.45 | 5.75 | 0.0538 | 0.9576 | 1797.20 | 5.78 | 0.0541 | 0.9530 | 1795.44 | 5.77 | 0.0540 |
| | $AD_{9\times60}$ | 0.2210 | 1341.07 | 4.49 | 0.0643 | 0.2020 | 1352.55 | 4.53 | 0.0643 | 0.1900 | 1370.06 | 4.58 | 0.0637 |
| | **QuadAttac**$K_{9\times60}$ | **1.0000** | **533.11** | **1.92** | **0.0430** | **0.9992** | **540.04** | **1.94** | **0.0433** | **0.9970** | **537.46** | **1.93** | **0.0431** |
| | **QuadAttac**$K_{9\times30}$ | 0.9950 | 1052.24 | 3.55 | 0.0471 | 0.9926 | 1052.56 | 3.55 | 0.0470 | 0.9910 | 1045.70 | 3.53 | 0.0471 |
| Top-5 | $CW^K_{1\times30}$ | 0.2100 | 1377.22 | 4.40 | 0.0413 | 0.1934 | 1378.10 | 4.40 | 0.0415 | 0.1800 | 1374.54 | 4.39 | 0.0414 |
| | $AD_{1\times30}$ | 0.8480 | 1351.05 | 4.32 | 0.0389 | 0.8324 | 1352.17 | 4.32 | 0.0390 | 0.8140 | 1357.48 | 4.34 | 0.0391 |
| | **QuadAttac**$K_{1\times30}$ | **0.9450** | **765.54** | **2.55** | **0.0289** | **0.9380** | **759.47** | **2.54** | **0.0289** | **0.9330** | **757.77** | **2.53** | **0.0289** |
| | $CW^K_{9\times30}$ | 0.4470 | 1003.94 | 3.28 | 0.0382 | 0.4216 | 1026.70 | 3.35 | 0.0384 | 0.4080 | 1042.19 | 3.40 | 0.0384 |
| | $AD_{9\times30}$ | 0.9550 | 549.73 | 1.91 | 0.0342 | 0.9498 | 557.15 | 1.93 | 0.0342 | 0.9380 | 553.08 | 1.92 | 0.0341 |
| | **QuadAttac**$K_{9\times30}$ | **0.9970** | **462.81** | **1.60** | **0.0277** | **0.9948** | **465.20** | **1.61** | **0.0278** | **0.9930** | **461.15** | **1.60** | **0.0277** |
| Top-1 | $CW^K_{1\times30}$ | **1.0000** | 483.86 | 1.53 | 0.0142 | 0.9978 | 483.88 | 1.53 | 0.0142 | 0.9960 | 485.16 | 1.54 | 0.0142 |
| | $AD_{1\times30}$ | **1.0000** | 465.62 | 1.48 | **0.0141** | **0.9990** | 467.67 | 1.48 | **0.0141** | 0.9980 | 469.54 | 1.49 | **0.0142** |
| | **QuadAttac**$K_{1\times30}$ | **1.0000** | **446.84** | **1.44** | 0.0143 | **0.9990** | **448.22** | **1.44** | 0.0143 | 0.9980 | **448.70** | **1.45** | 0.0144 |
| | | | | | | DenseNet-121 [Huang et al., 2017] | | | | | | | |
| Top-20 | **QuadAttac**$K_{1\times60}$ | 0.9310 | 2394.04 | 7.82 | 0.0907 | 0.9184 | 2388.03 | 7.80 | 0.0908 | 0.9070 | 2387.15 | 7.80 | 0.0908 |
| | **QuadAttac**$K_{1\times30}$ | 0.3280 | 2626.42 | 8.42 | 0.0793 | 0.3204 | 2630.12 | 8.43 | 0.0794 | 0.3160 | 2632.14 | 8.44 | 0.0795 |
| Top-15 | **QuadAttac**$K_{1\times60}$ | 0.9910 | 1880.34 | 6.17 | 0.0682 | 0.9846 | 1882.11 | 6.18 | 0.0683 | 0.9790 | 1874.28 | 6.15 | 0.0679 |
| | **QuadAttac**$K_{1\times30}$ | 0.9130 | 2176.90 | 6.98 | 0.0644 | 0.9072 | 2174.17 | 6.97 | 0.0641 | 0.9020 | 2167.97 | 6.95 | 0.0639 |
| Top-10 | $CW^K_{1\times60}$ | 0.1650 | 2088.38 | 6.74 | 0.0755 | 0.1388 | 2090.22 | 6.75 | 0.0750 | 0.1260 | 2082.13 | 6.73 | 0.0750 |
| | $AD_{1\times60}$ | 0.5200 | 1432.66 | 4.78 | 0.0650 | 0.5110 | 1426.53 | 4.76 | 0.0646 | 0.4920 | 1429.76 | 4.77 | 0.0641 |
| | **QuadAttac**$K_{1\times60}$ | **0.9980** | **1387.24** | **4.61** | **0.0495** | **0.9960** | **1395.19** | **4.63** | **0.0498** | **0.9930** | **1392.97** | **4.62** | **0.0497** |
| | **QuadAttac**$K_{1\times30}$ | 0.9930 | 1623.21 | 5.23 | 0.0471 | 0.9894 | 1626.75 | 5.24 | 0.0471 | 0.9870 | 1637.11 | 5.27 | 0.0470 |
| | $CW^K_{9\times60}$ | 0.4630 | 1902.17 | 6.19 | 0.0726 | 0.4470 | 1895.21 | 6.16 | 0.0724 | 0.4120 | 1902.44 | 6.18 | 0.0723 |
| | $AD_{9\times60}$ | 0.6830 | 1070.25 | 3.66 | 0.0588 | 0.6648 | 1073.88 | 3.67 | 0.0587 | 0.6460 | 1071.64 | 3.66 | 0.0587 |
| | **QuadAttac**$K_{9\times60}$ | **0.9990** | **451.30** | **1.61** | **0.0394** | **0.9984** | **443.78** | **1.59** | **0.0393** | **0.9970** | **446.99** | **1.60** | **0.0399** |
| | $AD_{9\times30}$ | 0.0940 | 1588.57 | 5.17 | 0.0581 | 0.0778 | 1572.15 | 5.12 | 0.0578 | 0.0690 | 1572.70 | 5.13 | 0.0580 |
| | **QuadAttac**$K_{9\times30}$ | **0.9990** | **838.91** | **2.84** | **0.0406** | **0.9956** | **841.88** | **2.85** | **0.0407** | **0.9930** | **843.84** | **2.86** | **0.0407** |
| Top-5 | $CW^K_{1\times30}$ | 0.5560 | 1062.48 | 3.45 | 0.0376 | 0.5358 | 1064.42 | 3.46 | 0.0377 | 0.5130 | 1057.52 | 3.44 | 0.0378 |
| | $AD_{1\times30}$ | 0.9120 | 716.61 | 2.46 | 0.0355 | 0.9040 | 714.59 | 2.45 | 0.0354 | 0.8890 | 710.44 | 2.44 | 0.0354 |
| | **QuadAttac**$K_{1\times30}$ | **0.9400** | **676.94** | **2.28** | **0.0269** | **0.9284** | **680.24** | **2.28** | **0.0268** | **0.9130** | **677.48** | **2.28** | **0.0267** |
| | $CW^K_{9\times30}$ | 0.8690 | 899.15 | 2.97 | 0.0363 | 0.8566 | 905.02 | 2.99 | 0.0364 | 0.8430 | 905.64 | 2.99 | 0.0364 |
| | $AD_{9\times30}$ | 0.9810 | 438.53 | 1.55 | 0.0326 | 0.9694 | 441.40 | 1.56 | 0.0329 | 0.9570 | 447.02 | 1.58 | 0.0332 |
| | **QuadAttac**$K_{9\times30}$ | **0.9980** | **427.58** | **1.47** | **0.0259** | **0.9932** | **431.84** | **1.49** | **0.0261** | **0.9900** | **424.47** | **1.46** | **0.0262** |
| Top-1 | $CW^K_{1\times30}$ | **1.0000** | 437.41 | 1.39 | **0.0134** | 0.9988 | 442.44 | 1.41 | **0.0135** | 0.9980 | 443.43 | 1.41 | **0.0135** |
| | $AD_{1\times30}$ | **1.0000** | 422.10 | 1.35 | 0.0137 | **0.9990** | 426.98 | 1.36 | 0.0137 | 0.9970 | 428.25 | 1.37 | 0.0137 |
| | **QuadAttac**$K_{1\times30}$ | **1.0000** | **377.99** | **1.25** | 0.0138 | **0.9990** | **383.13** | **1.26** | 0.0138 | 0.9970 | **386.19** | **1.27** | 0.0138 |
| | | | | | | ViT-B [Dosovitskiy et al., 2020] | | | | | | | |
| Top-20 | **QuadAttac**$K_{1\times60}$ | 0.6100 | 2651.86 | 8.68 | 0.0847 | 0.5964 | 2642.73 | 8.65 | 0.0841 | 0.5810 | 2640.33 | 8.65 | 0.0841 |
| Top-15 | **QuadAttac**$K_{1\times60}$ | 0.7150 | 1919.22 | 6.28 | 0.0618 | 0.7062 | 1920.08 | 6.28 | 0.0618 | 0.6950 | 1927.03 | 6.30 | 0.0620 |
| Top-10 | **QuadAttac**$K_{1\times60}$ | 0.8660 | 1582.15 | 5.09 | 0.0422 | 0.8480 | 1579.52 | 5.08 | 0.0421 | 0.8230 | 1595.84 | 5.13 | 0.0420 |
| | $AD_{9\times60}$ | 0.0900 | 1361.59 | 4.45 | 0.0472 | 0.0690 | 1398.54 | 4.56 | 0.0474 | 0.0600 | 1380.14 | 4.52 | 0.0476 |
| | **QuadAttac**$K_{9\times60}$ | **0.9850** | **991.22** | **3.36** | **0.0383** | **0.9802** | **978.89** | **3.33** | **0.0383** | **0.9730** | **979.63** | **3.33** | **0.0383** |
| | **QuadAttac**$K_{9\times30}$ | 0.0590 | 1200.33 | 3.80 | 0.0295 | 0.0550 | 1193.83 | 3.78 | 0.0296 | 0.0470 | 1156.53 | 3.68 | 0.0294 |
| Top-5 | $AD_{1\times30}$ | 0.2650 | 1076.10 | 3.42 | 0.0286 | 0.2450 | 1079.95 | 3.43 | 0.0285 | 0.2320 | 1089.83 | 3.46 | 0.0285 |
| | **QuadAttac**$K_{1\times30}$ | **0.4960** | **1060.31** | **3.35** | **0.0267** | **0.4692** | **1062.17** | **3.35** | **0.0267** | **0.4380** | **1060.54** | **3.35** | **0.0267** |
| | $CW^K_{9\times30}$ | 0.1110 | 999.43 | 3.18 | 0.0278 | 0.1070 | 984.00 | 3.14 | 0.0277 | 0.0970 | 963.58 | 3.08 | 0.0275 |
| | $AD_{9\times30}$ | 0.4760 | 947.39 | 3.06 | 0.0282 | 0.4580 | 950.24 | 3.07 | 0.0282 | 0.4320 | 945.25 | 3.05 | 0.0282 |
| | **QuadAttac**$K_{9\times30}$ | **0.7670** | **910.88** | **2.95** | **0.0264** | **0.7530** | **922.78** | **2.98** | **0.0264** | **0.7390** | **945.03** | **3.04** | **0.0266** |
| Top-1 | $CW^K_{1\times30}$ | **0.9940** | 418.85 | **1.44** | **0.0162** | **0.9914** | 420.89 | 1.45 | **0.0163** | **0.9890** | 423.64 | 1.46 | **0.0164** |
| | $AD_{1\times30}$ | 0.9910 | **418.44** | 1.46 | 0.0176 | 0.9898 | **414.87** | **1.45** | 0.0176 | 0.9870 | **410.74** | **1.43** | 0.0174 |
| | **QuadAttac**$K_{1\times30}$ | 0.9910 | 451.55 | 1.53 | 0.0169 | 0.9884 | 444.51 | 1.52 | 0.0169 | 0.9840 | 434.08 | 1.48 | 0.0167 |
| | | | | | | DeiT-S [Touvron et al., 2021] | | | | | | | |
| Top-20 | **QuadAttac**$K_{1\times60}$ | 0.7670 | 2164.35 | 7.02 | 0.0707 | 0.7496 | 2170.06 | 7.04 | 0.0711 | 0.7320 | 2179.99 | 7.08 | 0.0713 |
| Top-15 | **QuadAttac**$K_{1\times60}$ | 0.9400 | 1710.90 | 5.55 | 0.0537 | 0.9308 | 1712.43 | 5.55 | 0.0535 | 0.9170 | 1706.33 | 5.53 | 0.0534 |
| | **QuadAttac**$K_{1\times30}$ | 0.0490 | 1414.01 | 4.50 | 0.0379 | 0.0444 | 1416.91 | 4.51 | 0.0381 | 0.0410 | 1427.73 | 4.54 | 0.0379 |
| Top-10 | $AD_{1\times60}$ | 0.0610 | 1247.46 | 4.05 | 0.0430 | 0.0534 | 1224.28 | 3.97 | 0.0419 | 0.0480 | 1225.32 | 3.97 | 0.0421 |
| | **QuadAttac**$K_{1\times60}$ | **0.9800** | **1161.60** | **3.75** | **0.0356** | **0.9778** | **1163.31** | **3.76** | **0.0358** | **0.9750** | **1168.52** | **3.78** | **0.0358** |
| | **QuadAttac**$K_{1\times30}$ | 0.2000 | 1076.08 | 3.41 | 0.0278 | 0.1958 | 1067.14 | 3.38 | 0.0275 | 0.1920 | 1062.29 | 3.36 | 0.0273 |
| | $CW^K_{9\times60}$ | 0.0390 | 1000.92 | 3.24 | 0.0360 | 0.0334 | 1000.35 | 3.23 | 0.0352 | 0.0300 | 987.21 | 3.19 | 0.0350 |
| | $AD_{9\times60}$ | 0.1250 | 812.31 | 2.71 | 0.0356 | 0.1122 | 829.99 | 2.76 | 0.0356 | 0.1000 | 819.51 | 2.73 | 0.0356 |
| | **QuadAttac**$K_{9\times60}$ | **0.9990** | **546.66** | **1.95** | **0.0294** | **0.9978** | **543.50** | **1.95** | **0.0296** | **0.9950** | **542.95** | **1.95** | **0.0298** |
| | **QuadAttac**$K_{9\times30}$ | 0.4410 | 1019.14 | 3.25 | 0.0274 | 0.4232 | 1033.34 | 3.29 | 0.0277 | 0.4090 | 1027.87 | 3.27 | 0.0275 |
| Top-5 | $CW^K_{1\times30}$ | 0.0730 | 924.23 | 2.94 | 0.0266 | 0.0670 | 913.38 | 2.91 | 0.0265 | 0.0620 | 903.37 | 2.88 | 0.0263 |
| | $AD_{1\times30}$ | 0.4830 | 1010.38 | 3.20 | 0.0271 | 0.4640 | 1016.17 | 3.22 | 0.0272 | 0.4400 | 1019.77 | 3.23 | 0.0273 |
| | **QuadAttac**$K_{1\times30}$ | **0.8250** | **849.16** | **2.72** | **0.0246** | **0.8068** | **843.14** | **2.71** | **0.0244** | **0.8000** | **846.60** | **2.72** | **0.0245** |
| | $CW^K_{9\times30}$ | 0.2310 | 909.93 | 2.91 | 0.0268 | 0.2204 | 902.41 | 2.88 | 0.0266 | 0.2160 | 898.71 | 2.87 | 0.0264 |
| | $AD_{9\times30}$ | 0.7510 | 858.41 | 2.78 | 0.0267 | 0.7270 | 836.38 | 2.71 | 0.0264 | 0.7080 | 836.42 | 2.72 | 0.0265 |
| | **QuadAttac**$K_{9\times30}$ | **0.9550** | **592.60** | **2.00** | **0.0229** | **0.9528** | **579.32** | **1.96** | **0.0227** | **0.9500** | **573.48** | **1.94** | **0.0226** |
| Top-1 | $CW^K_{1\times30}$ | 0.9990 | 417.62 | 1.39 | **0.0149** | 0.9976 | 415.49 | 1.38 | **0.0148** | **0.9970** | 410.32 | 1.36 | **0.0147** |
| | $AD_{1\times30}$ | **1.0000** | 407.16 | 1.37 | 0.0158 | **0.9982** | 404.24 | 1.36 | 0.0157 | **0.9970** | 397.79 | 1.34 | 0.0156 |
| | **QuadAttac**$K_{1\times30}$ | 0.9960 | **294.88** | **1.08** | 0.0153 | 0.9936 | **295.00** | **1.08** | 0.0153 | 0.9900 | **297.33** | **1.09** | 0.0154 |

Table 1: Comparisons under the ordered top-$K$ targeted attack protocol using randomly selected and ordered $K$ targets (GT exclusive) in ImageNet using four popular models (ResNet-50, DenseNet-121, ViT-B and DEiT-S). The $CW^K$ and AD methods are proposed in [Zhang and Wu, 2020]. We test $9 \times 30$ for $K = 5$, and $9 \times \{30, 60\}$ for $K = 10$. For each protocol (e.g., top-20) and each budget (e.g., $1\times60$), if an attack method (e.g., $CW^K$ or AD) is not listed, it means that it fails completely, i.e., zero ASR, and thus is ommitted in the table for clarity.

**Qualitative Results.** We show more examples in Appendix B. From those, we note that for our QuadAttac$K$, when the target classes deviate significantly from the original predicted classes, we often observe a perturbation that achieves the prescribed top-$K$ targets without a substantial margin between each of the top-$K$ targets. This outcome reflects the desirable effect of our approach, as the primary objective of the ordered top-$K$ attack is to enforce a specific class order rather than optimizing for class probability differences. Our method's strength lies in its ability to handle such scenarios without relying on explicit assumptions about the distances between classes. By prioritizing the order constraints, our QuadAttac$K$ offers a robust solution that aligns with the fundamental goal of enforcing class order in adversarial attacks.

## 5 Limitations of Our QuadAttac$K$

There are two main limitations that worth further exploring. Our proposed QuadAttac$K$ is specifically designed for clear-box attack setting, which makes it not directly applicable to opaque-box attacks. The attacks learned by our QuadAttack$K$ are not easily transferable between different networks, as they are specifically optimized in the feature embedding space for the target model. Exploring the transferability of attacks and developing more generalizable strategies across various networks could be an intriguing direction for future research. In addition, our QuadAttac$K$ entails solving a QP at each iteration, which introduces additional computational overhead compared to methods like CW$^K$ and AD [Zhang and Wu, 2020]. For example, for ResNet-50, we observed an average QuadAttac$K$ performs 2.47 attack iterations per second whereas AD performs 32.02 iterations per second (a factor of 12.96). For ViT-B, QuadAtta$K$ performs 2.96 attack iterations per second whereas AD performs 11.86 iterations per second (a factor of 4). We note that as the target model becomes larger, the adversarial loss constitutes a smaller fraction of total runtime thus the ratio tends toward one. Further, we note that quicker attack iterations of QuadAttac$K$ on ViT-B which indicate our QP solver converges faster on ViT-B attacks. To address the overhead of our QuadAttac$K$, we will explore and compare how the QP solver could be adjusted to initialize the QP solver at the previous iteration's solution to nearly eradicate the cost of the QP solver in future work.

## 6 Broader Impact

Our proposed QuadAttac$K$ method showcases the effectiveness of utilizing QP techniques in the challenging domain of learning ordered top-$K$ clear-box targeted adversarial attacks. The underlying QP formulation offers opportunities for exploring other applications beyond adversarial attacks. For instance, it could be leveraged to design new loss functions going beyond the traditional cross-entropy loss or the label smoothing variant [Szegedy et al., 2015], and thus jointly optimizing accuracy and robustness. We can enforce semantically meaningful class orders in training a network from scratch, thus allowing for the incorporation of explicit constraints in neural network predictions and potentially resulting in a more interpretable and controlled decision-making process.

*Potential Negative Societal Impact.* As discussed in the introduction, there are some potential scenarios in practice for which the proposed ordered top-$K$ adversarial attacks may be risky if applied. However, since we focus on clear-box attacks, they are less directly applicable in practice compared to opaque-box attacks, which makes the concern less serious.

## 7 Conclusions

This paper presents a quadratic programming (QP) based method for learning ordered top-$K$ clear-box targeted attacks. By formulating the task as a constrained optimization problem, we demonstrate the capability to achieve successful attacks with larger values of $K$ ($K > 10$) compared to previous methods. The proposed QuadAttac$K$ is tested in the ImageNet-1k classification using ResNet-50 and DenseNet-121, and ViT-B and DEiT-S. It successfully pushes the boundary of successful ordered top-$K$ attacks from $K = 10$ up to $K = 20$ at a cheap budget ($1 \times 60$) and further improves attack success rates for $K = 5$ for all tested models, while retaining the performance for $K = 1$. The promising results highlight the potential of QP and constrained optimization as powerful tools opening new avenues for research in adversarial attacks and beyond.

## Acknowledgements

This research is partly supported by ARO Grant W911NF1810295, ARO Grant W911NF2210010, NSF IIS-1909644, NSF CMMI-2024688 and NSF IUSE-2013451. The views and conclusions contained herein are those of the authors and should not be interpreted as necessarily representing the official policies or endorsements, either expressed or implied, of the ARO, NSF, or the U.S. Government. The U.S. Government is authorized to reproduce and distribute reprints for Governmental purposes not withstanding any copyright annotation thereon.

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

# A Experimental Settings

The selection of learning rates $\gamma$ and $\lambda$ values (see Eqn. 10) in the context of learning attacks requires careful consideration to achieve the desired trade-offs (see Fig. 3) and optimize the attack performance. In this regard, the learning rate is adapted based on the value of the number of targets, $K$. Empirically, we observed that as $K$ increases, a higher learning rate tends to yield a better balance between Attack Success Rate (ASR) and perturbation energy consumption. Specifically, for different ranges of $K$, the learning rate is adjusted accordingly, ensuring an appropriate scaling factor to guide the optimization process. Additionally, we find convolutional models require a larger learning rate than Transformer models to reach desirable attacks in all tested methods.

- If $K < 5$, the learning rate is set to $\gamma = 0.75e - 3$ (for all the four models).
- If $5 \leq K < 10$, the learning rate is set to $\gamma = 1.0e - 3$ (for ViT-B and DEiT-S) and $\gamma = 2.0e - 3$ (for ResNet-50 and DenseNet-121).
- If $10 \leq K < 15$, the learning rate is set to $\gamma = 1.0e - 3$ (for ViT-B and DEiT-S) and $\gamma = 3e - 3$ (for ResNet-50 and DenseNet-121).
- If $15 \leq K < 20$, the learning rate is set to $\gamma = 1.5e - 3$ (for ViT-B and DEiT-S) and $\gamma = 3.5e - 3$ (for ResNet-50 and DenseNet-121).
- If $K \geq 20$, the learning rate is set to $\gamma = 2e - 3$ (for ViT-B and DEiT-S) and $\gamma = 4e - 3$ (for ResNet-50 and DenseNet-121).

Similarly, the choice of the weight parameter $\lambda$ in the loss function (Eqn. 10) also plays a crucial role. We use $\lambda$ to weight the first term in both Eqn. 3 and Eqn. 10 (the loss term that finds a successful attack when optimized) and we leave the second term (the energy penalty) unweighted. In challenging attacks where $K \geq 5$, the range of suitable $\lambda$ values that achieve desirable trade-offs between ASR and energy is significantly wider compared to easier attacks ($K = 1$). Given the multitude of appropriate $\lambda$ values for difficult attacks, we do not perform explicit tuning of $\lambda$ in computing results in Table 1, since different choices of $\lambda$ would correspond to different points along the energy/ASR trade-off curve (which are used in generating the trade-off curves in Fig 3).

For attacks with $K = 1$, selecting an excessively high $\lambda$ can lead to inefficient energy usage. For instance, consider an attack with $\lambda = 5$ achieving an ASR of 1.0 and an energy cost of 2.0, while increasing $\lambda$ to 10 maintains the ASR at 1.0 but raises the energy cost to 5.0. In this case, the choice of $\lambda = 5$ is preferable as it achieves the desired ASR with lower energy consumption.

For our QuadAttac$K$ which operates in a high-dimensional latent space with much higher loss magnitudes than CW$^K$ and $AD$ (which operate in the logit or probability space), a lower value of $\lambda$ is necessary to reach the optimal point for $K = 1$. Hence, we set $\lambda = 0.5$ for QuadAttac$K$ and $\lambda = 5$ for the logit/probability space losses for the $K = 1$ case. For all other values of $K$, we use $\lambda = 10$ since these attacks are more challenging and require a higher weight on the top-$K$ term to attain the desired ASR and none of the tested methods reach ASR saturation points on $K >= 5$ for the chosen $\lambda$.

# B More Qualitative Results

In addition to quantitative evaluations in Table 1, we provide detailed visualizations and qualitative analysis to gain deeper insights into the behavior and impact of our QuadAttac$K$ method on the classification models. These visualizations offer a comprehensive understanding of the attack process, showcasing the changes in both image perturbations and attention maps (for Transformer models). By examining the visual patterns and comparing the distributions of clean and attacked class predictions, we can explore the effects of our QuadAttac$K$ on the attacked models' predictions and gain valuable insights into the robustness of these models. These visual analyses serve as a valuable complement to our quantitative assessments, providing a holistic perspective on the performance and behavior of our QuadAttac$K$ across various attack scenarios.

Furthermore, it is important to note that while our QuadAttac$K$ does not prioritize maximizing the margins between class probabilities, its quadratic programming approach fundamentally enables the

| Clean Image & Attn | Adv. Example | Adv. Perturbation | Adv. Attention |
|---|---|---|---|

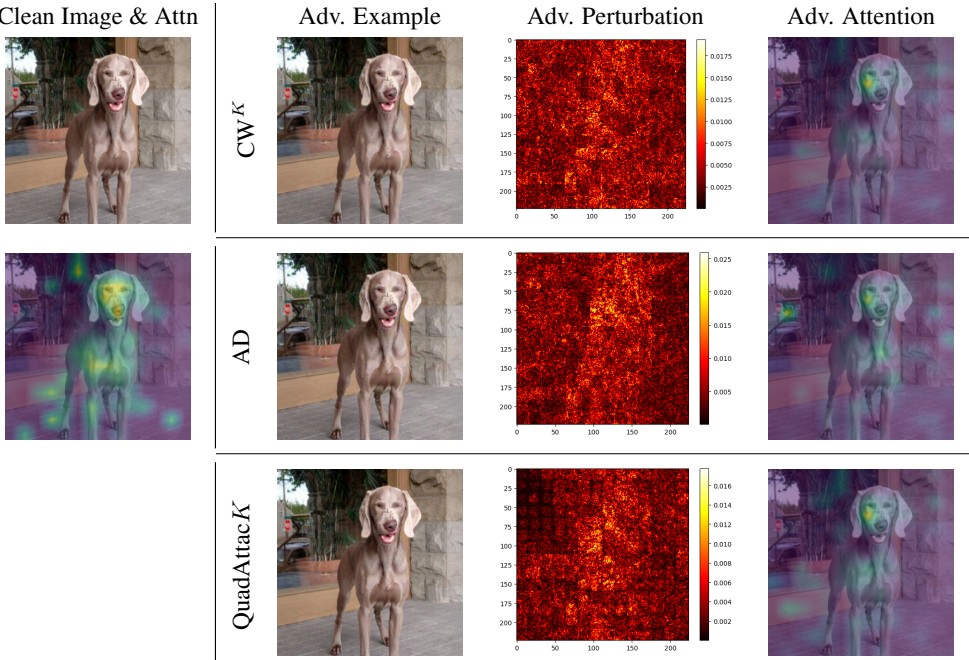

Figure 4: Examples of attacking DeIT-S [Touvron et al., 2021] with $K = 5$: the 1st and 2nd rows show results for $CW^K$ and Adversarial Distillation (AD) [Zhang and Wu, 2020] respectively, and the 3rd row shows results for QuadAttac$K$. The ground-truth label is *Weimaraner*. **The ordered top-5 targets** (randomly sampled and kept unchanged for the different attack methods) are: [`table lamp, langur, wig, hip, piggy bank`], while for example the original top-5 predictions of DEiT-S are [`Weimaraner, Bedlington terrier, German short-haired pointer, Great Dane, Yorkshire terrier`].

integration of such constraints if desired for other applications. The flexibility of our method allows for the incorporation of additional objectives or constraints that prioritize class separability, enabling researchers to tailor the optimization process according to specific needs. This versatility opens up avenues for exploring variations of the QuadAttac$K$ framework and adapting it to diverse scenarios where increasing the margins between class probabilities is a desirable objective.

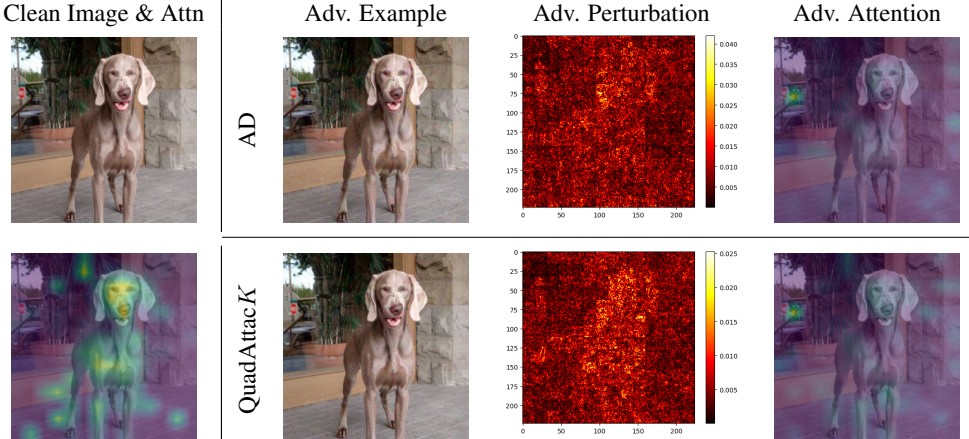

Figure 5: Examples of attacking DEiT-S [Touvron et al., 2021] with $K = 10$: the 1st row shows results for Adversarial Distillation (AD) [Zhang and Wu, 2020], and the 2rd row shows results for our QuadAttac$K$, and CW$^K$ fails for this example. The ground-truth label is *Weimaraner*. **The ordered top-10 targets** (randomly sampled and kept unchanged for the different attack methods) are: [`table lamp, langur, wig, hip, piggy bank, American Staffordshire terrier, school bus, crossword puzzle, entertainment center, ibex`], while for example the original top-10 predictions of DEiT-S are [`Weimaraner, Bedlington terrier, German short-haired pointer, Great Dane, Yorkshire terrier, Siamese cat, butcher shop, silky terrier, Italian greyhound, vizsla`].

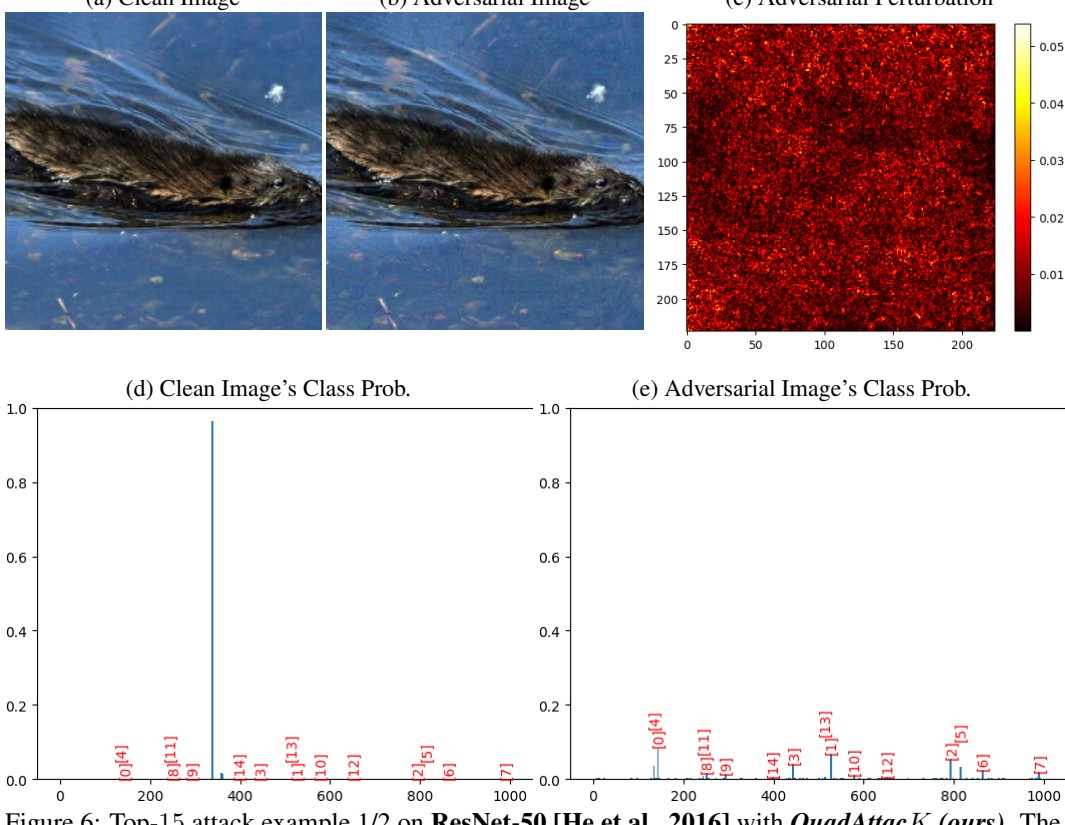

Figure 6: Top-15 attack example 1/2 on **ResNet-50 [He et al., 2016]** with *QuadAttac K (ours)*. The original Top-15 predictions are [ `beaver, mink, otter, weasel, platypus, porcupine, American coot, water snake, red-breasted merganser, guinea pig, sea lion, red-backed sandpiper, European gallinule, little blue heron, limpkin` ]. **The ordered Top-15 targets** (randomly sampled) are: [ `oystercatcher, desktop computer, shovel, bib, crane, spatula, torch, acorn, dalmatian, cheetah, gown, bull mastiff, microwave, cornet, puffer` ].

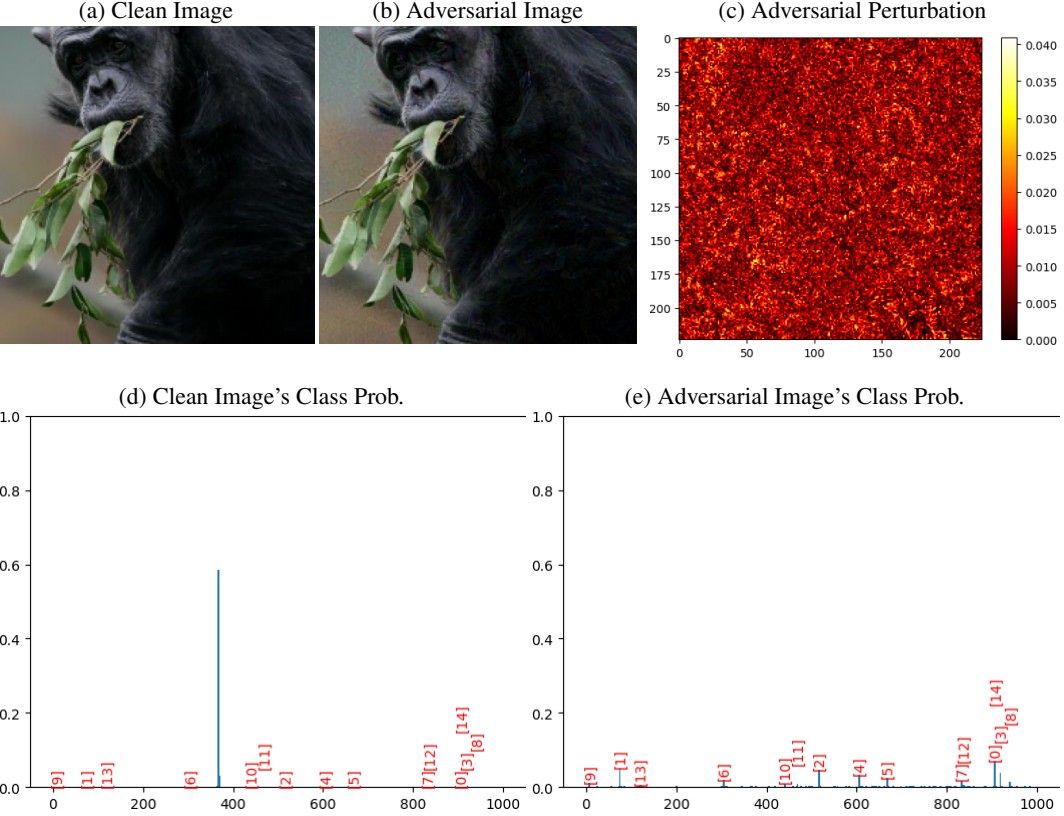

Figure 7: Top-15 attack example 1/2 on **DenseNet-121 [Huang et al., 2017]** with *QuadAttac K (ours)*. The original Top-15 predictions are [ chimpanzee, gorilla, siamang, orangutan, gibbon, sloth bear, colobus, howler monkey, guenon, American black bear, spider monkey, baboon, macaque, patas, water buffalo ]. **The ordered Top-15 targets** (randomly sampled) are: [ window shade, garden spider, cowboy hat, comic book, iPod, mortarboard, leaf beetle, stupa, zucchini, cock, beer bottle, cab, sunglass, fiddler crab, Windsor tie ].

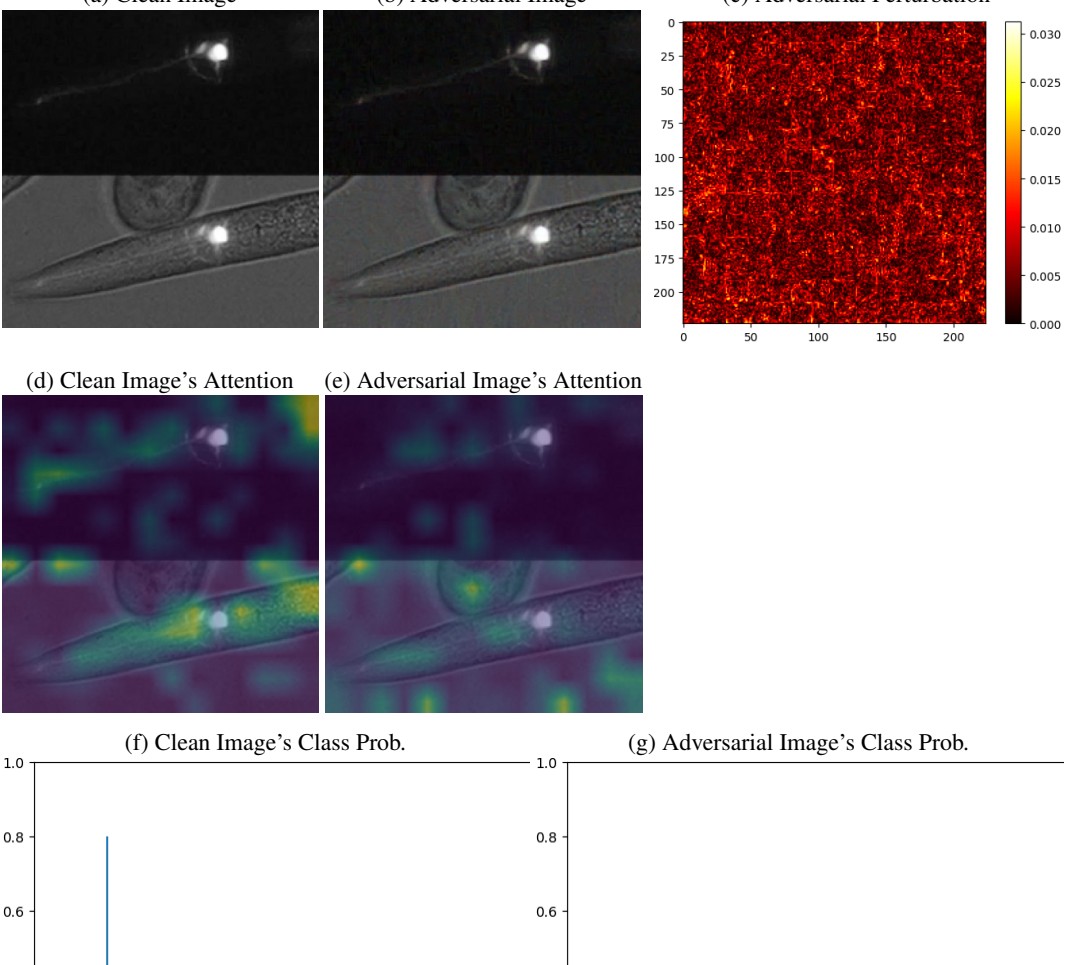

Figure 8: Top-15 attack example 1/2 on **DeiT-S [Touvron et al., 2021]** with *QuadAttac K (ours)*. The original Top-15 predictions are [ nematode, isopod, digital clock, jellyfish, crossword puzzle, flatworm, drumstick, jack-o'-lantern, knot, bassoon, safety pin, paper towel, thunder snake, matchstick, trombone ]. **The ordered Top-15 targets** (randomly sampled) are: [ axolotl, quail, hyena, carbonara, hen, oboe, mud turtle, robin, Italian greyhound, oystercatcher, space shuttle, airliner, Bedlington terrier, miniature pinscher, iron ].

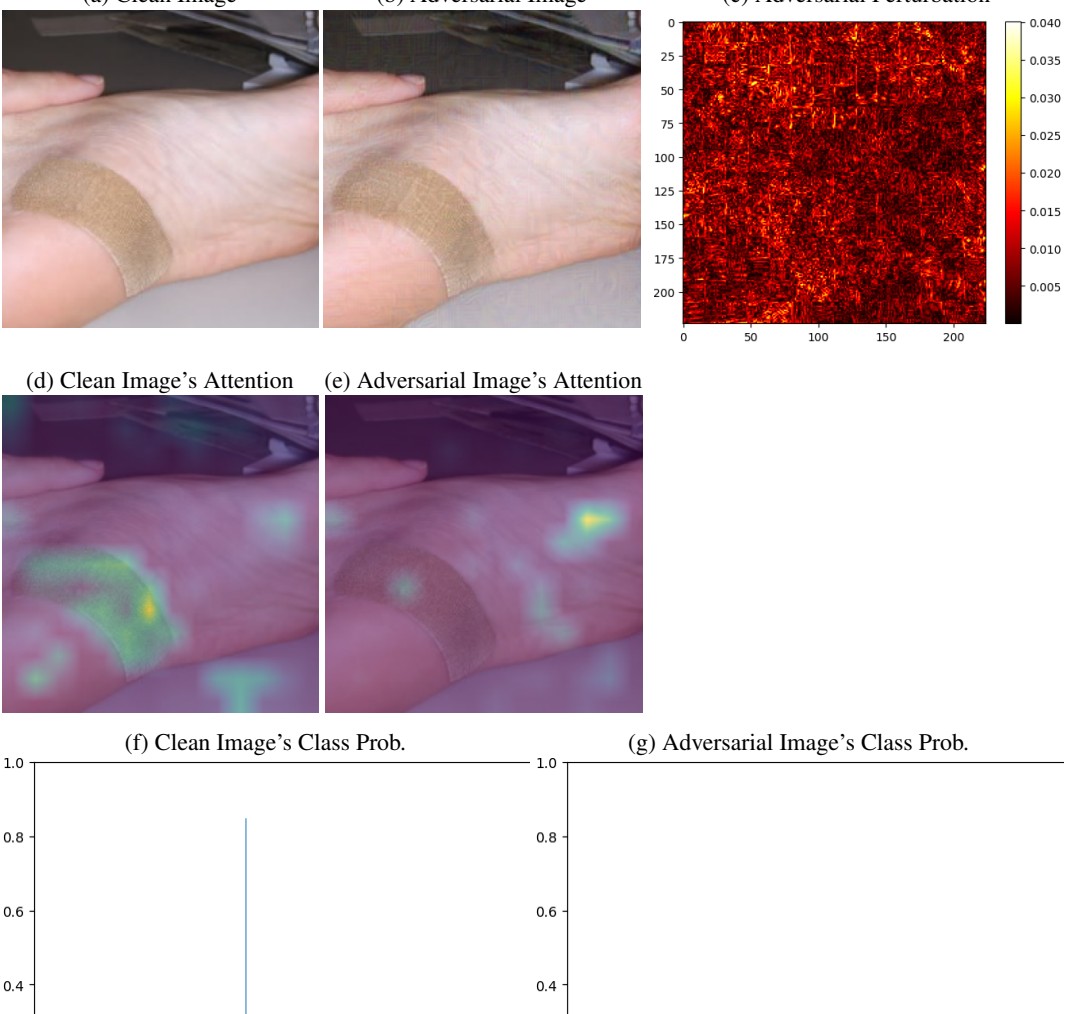

Figure 9: Top-15 attack example 1/2 on **ViT-B [Dosovitskiy et al., 2020]** with *QuadAttacK (ours)*. The original Top-15 predictions are [ Band Aid, airship, warplane, airliner, wing, mouse, space bar, face powder, missile, ballpoint, space shuttle, kite, modem, revolver, speedboat ]. **The ordered Top-15 targets** (randomly sampled) are: [ padlock, king snake, screwdriver, Welsh springer spaniel, bookshop, triumphal arch, shoe shop, Italian greyhound, diamondback, missile, drilling platform, worm fence, sea snake, African elephant, joystick ].

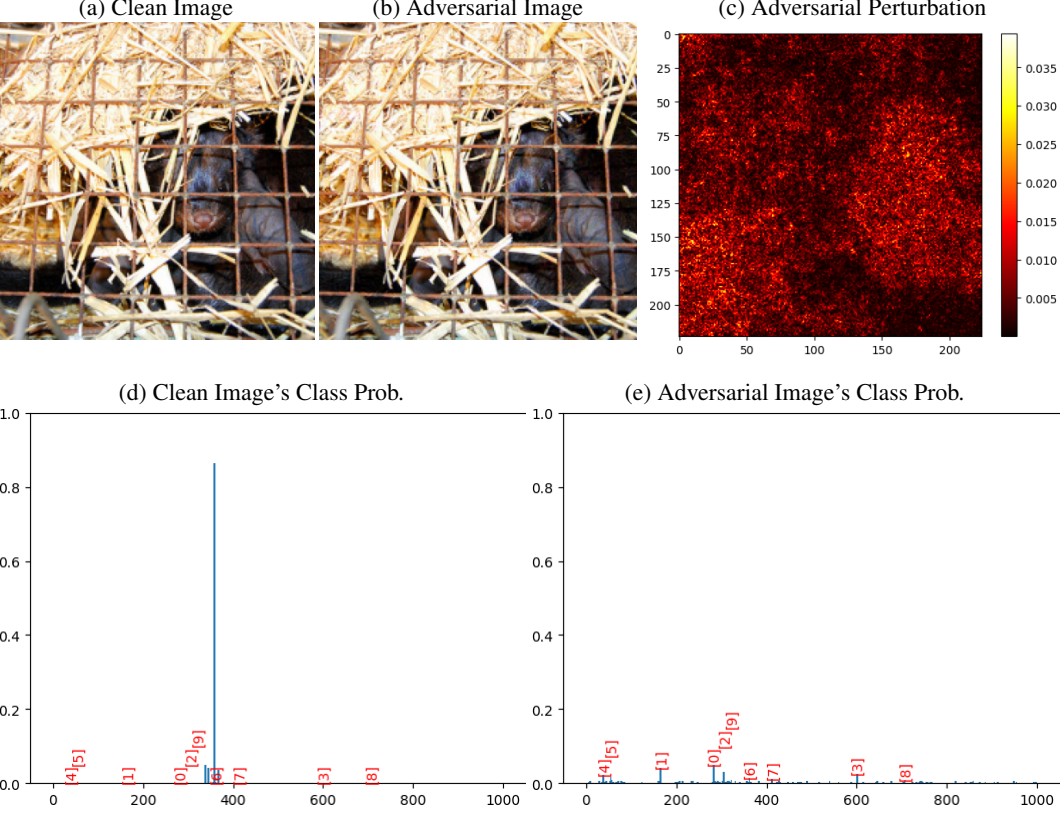

Figure 10: Top-10 attack example 1/2 on **ResNet-50 [He et al., 2016]** with *QuadAttacK (ours)*. The original Top-10 predictions are [ mink, beaver, hippopotamus, chimpanzee, weasel, otter, American alligator, mud turtle, Rottweiler, terrapin ]. **The ordered Top-10 targets** (randomly sampled) are: [ tiger cat, black-and-tan coonhound, dung beetle, hook, banded gecko, hognose snake, skunk, ashcan, patio, admiral ].

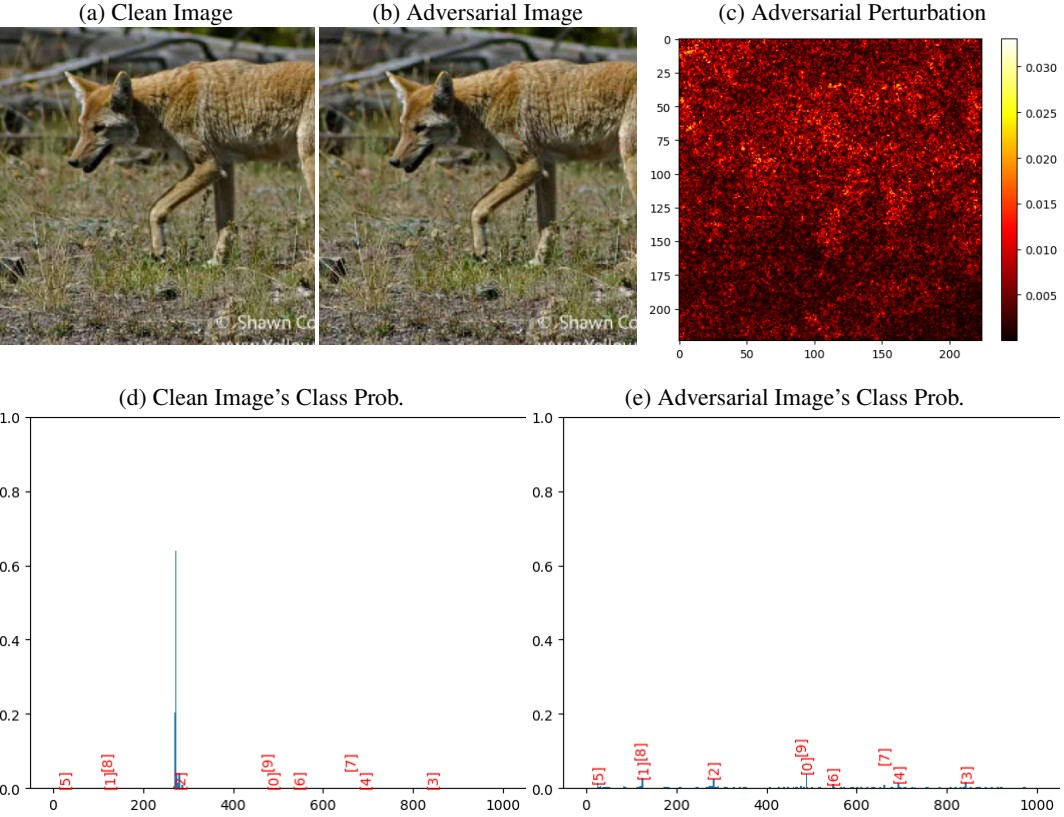

Figure 11: Top-10 attack example 1/2 on **DenseNet-121 [Huang et al., 2017]** with *QuadAttacK (ours)*. The original Top-10 predictions are [ `coyote, red wolf, grey fox, dhole, dingo, red fox, kit fox, lynx, timber wolf, hyena` ]. **The ordered Top-10 targets** (randomly sampled) are: [ `chain, hermit crab, tiger cat, sweatshirt, packet, European fire salamander, electric locomotive, mobile home, fiddler crab, car mirror` ].

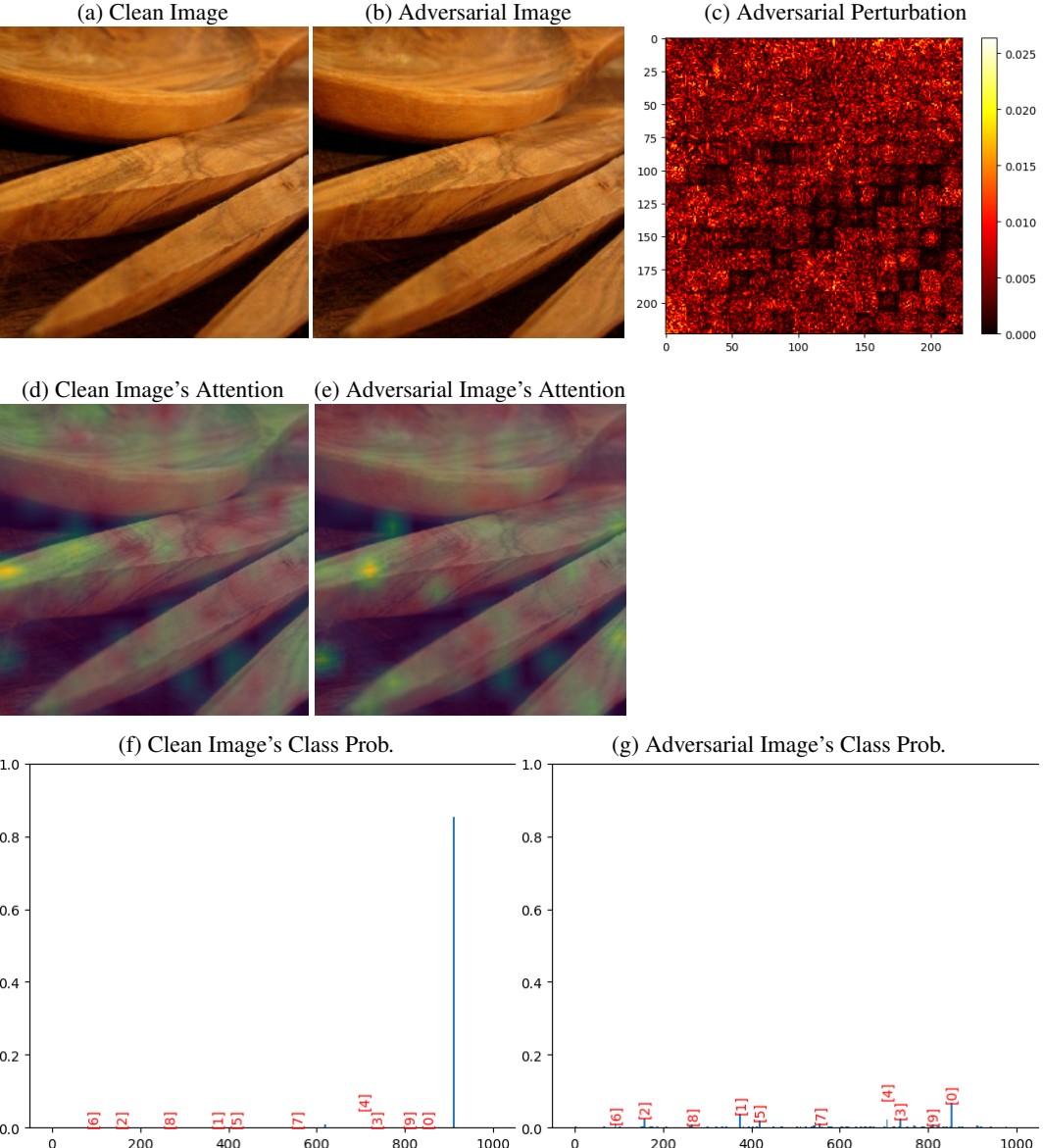

Figure 12: Top-10 attack example 1/2 on **DeiT-S [Touvron et al., 2021]** with *QuadAttacK (ours)*. The original Top-10 predictions are [ wooden spoon, ladle, French horn, snowmobile, mortar, spatula, paddle, rocking chair, fly, frying pan ]. **The ordered Top-10 targets** (randomly sampled) are: [ tennis ball, langur, toy terrier, pool table, patio, ballpoint, bee eater, fireboat, toy poodle, soup bowl ].

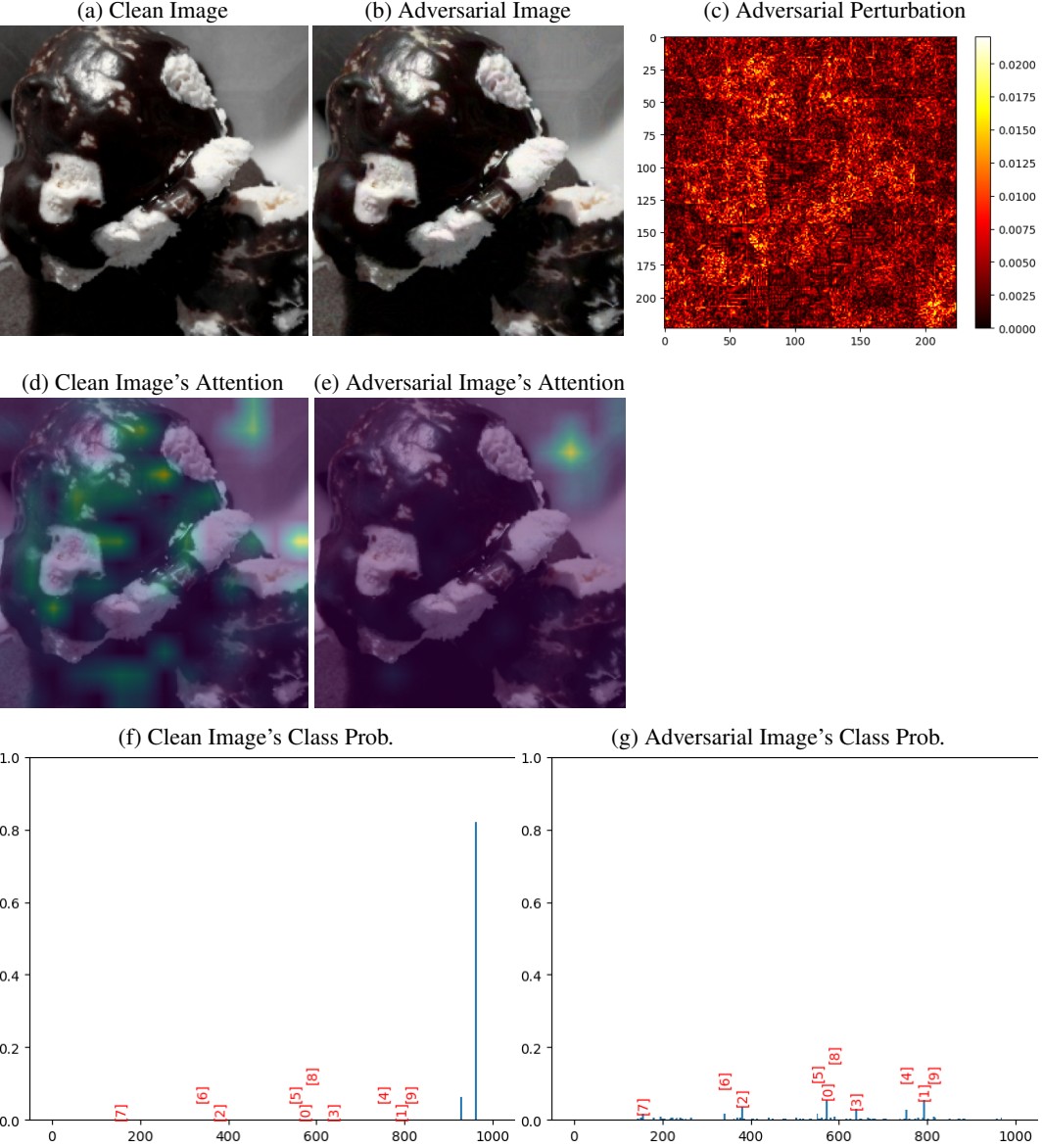

Figure 13: Top-10 attack example 1/2 on **ViT-B [Dosovitskiy et al., 2020]** with *QuadAttacK (ours)*. The original Top-10 predictions are [ chocolate sauce, ice cream, trifle, pomegranate, bakery, fig, hay, strawberry, burrito, rapeseed ]. **The ordered Top-10 targets** (randomly sampled) are: [ goblet, shopping cart, titi, maillot, racer, espresso maker, zebra, Shih-Tzu, hand blower, speedboat ].

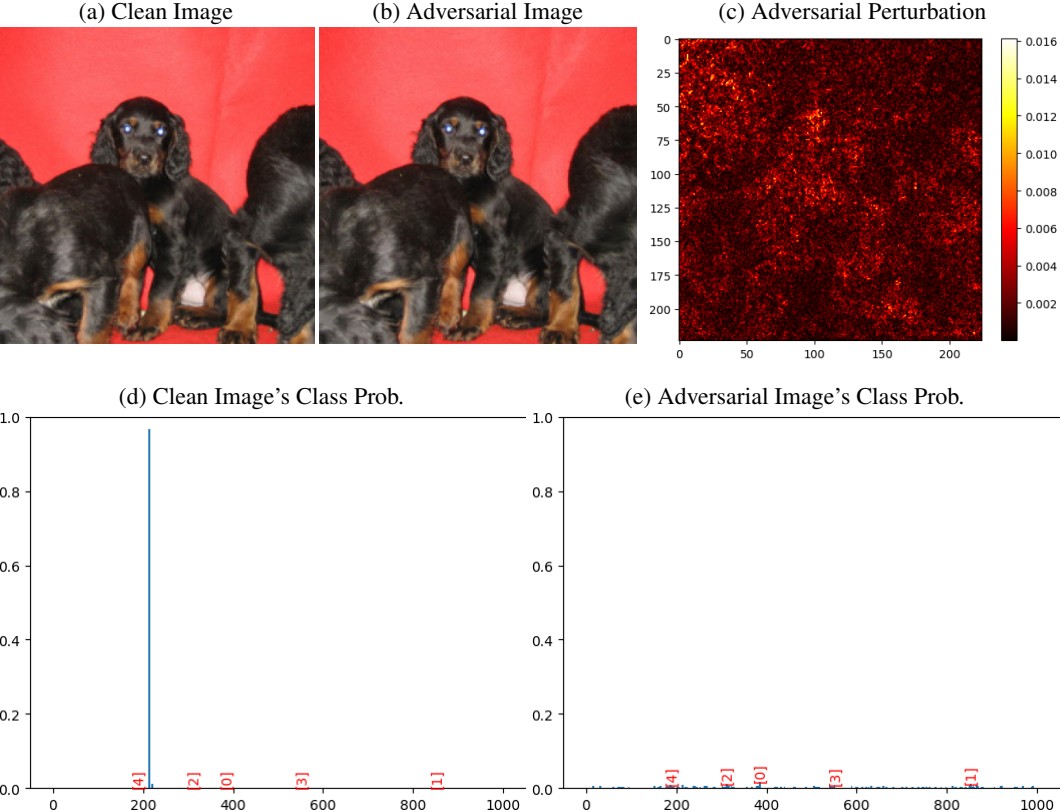

Figure 14: Top-5 attack example 1/2 on **ResNet-50 [He et al., 2016]** with *QuadAttac K (ours)*. The original Top-5 predictions are [ Gordon setter, cocker spaniel, Irish setter, Sussex spaniel, English setter ]. **The ordered Top-5 targets** (randomly sampled) are: [ indri, television, grasshopper, espresso maker, Yorkshire terrier ].

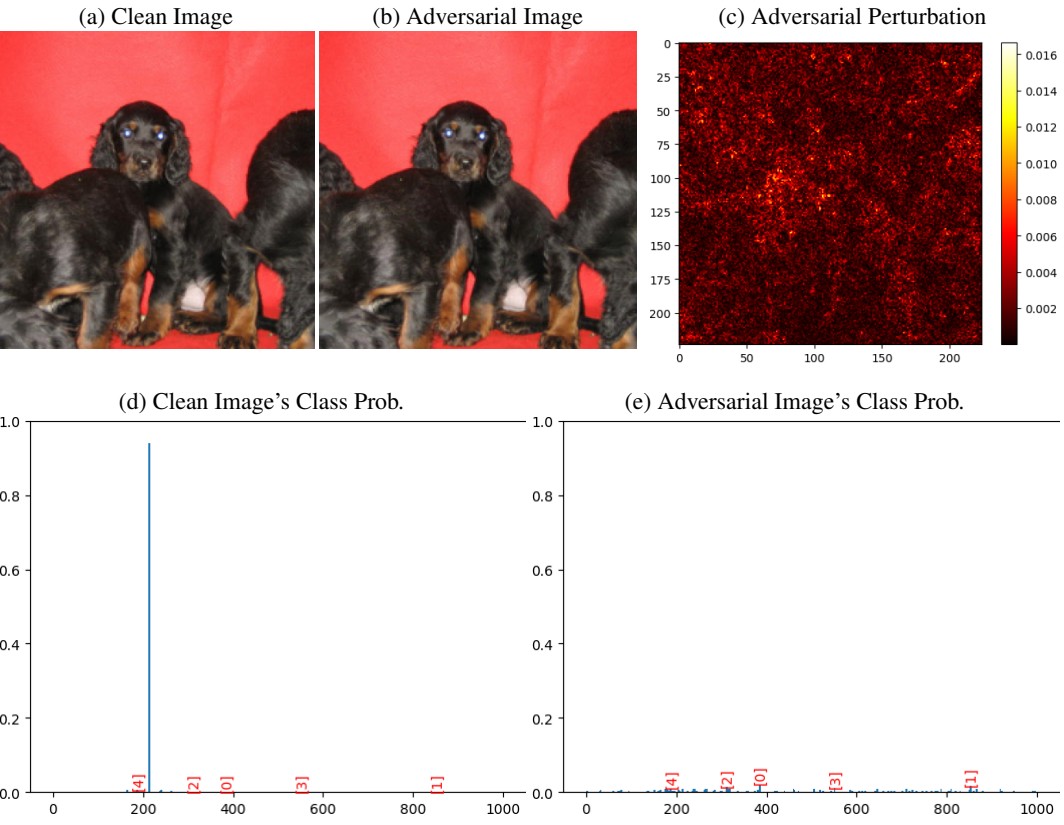

Figure 15: Top-5 attack example 1/2 on **DenseNet-121 [Huang et al., 2017]** with *QuadAttacK (ours)*. The original Top-5 predictions are [ Gordon setter, Yorkshire terrier, Bernese mountain dog, black-and-tan coonhound, Brabancon griffon ]. **The ordered Top-5 targets** (randomly sampled) are: [ indri, television, grasshopper, espresso maker, Yorkshire terrier ].

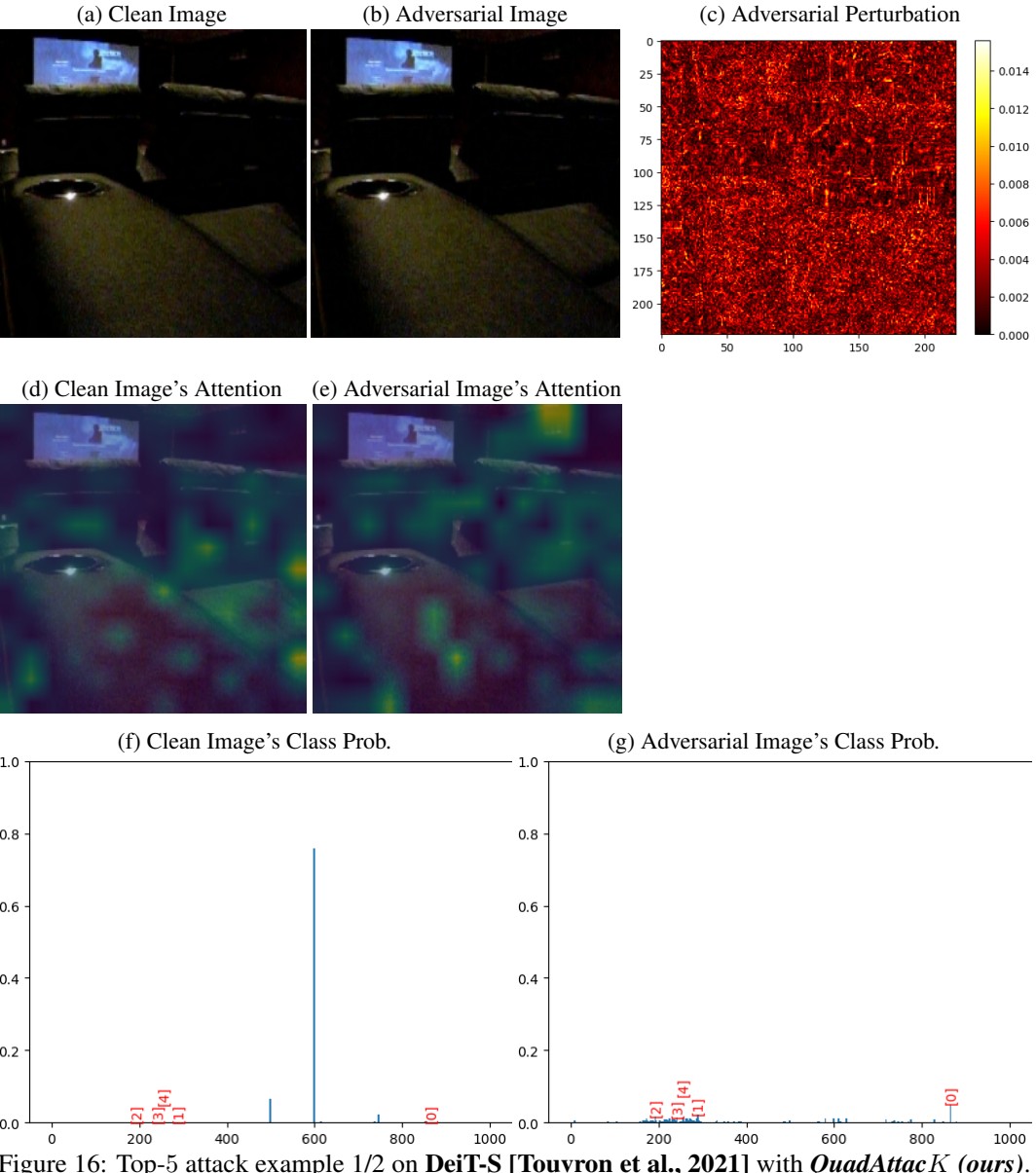

Figure 16: Top-5 attack example 1/2 on **DeiT-S [Touvron et al., 2021]** with *QuadAttacK (ours)*. The original Top-5 predictions are [ home theater, cinema, projector, theater curtain, pool table ]. **The ordered Top-5 targets** (randomly sampled) are: [ tow truck, snow leopard, cairn, EntleBucher, Leonberg ].

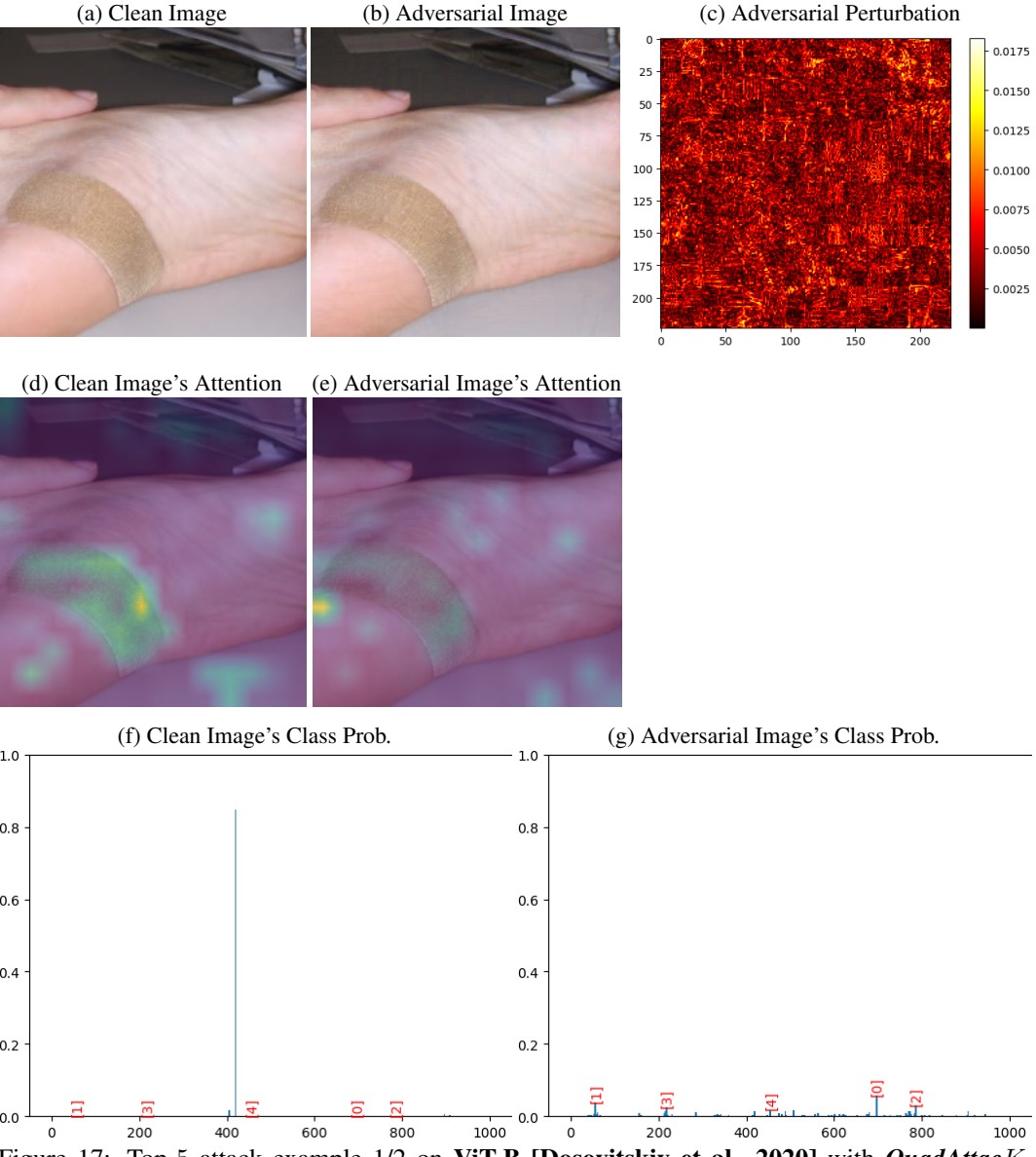

Figure 17: Top-5 attack example 1/2 on **ViT-B [Dosovitskiy et al., 2020]** with *QuadAttacK (ours)*. The original Top-5 predictions are [ Band Aid, airship, warplane, airliner, wing ]. **The ordered Top-5 targets** (randomly sampled) are: [ padlock, king snake, screwdriver, Welsh springer spaniel, bookshop  ].

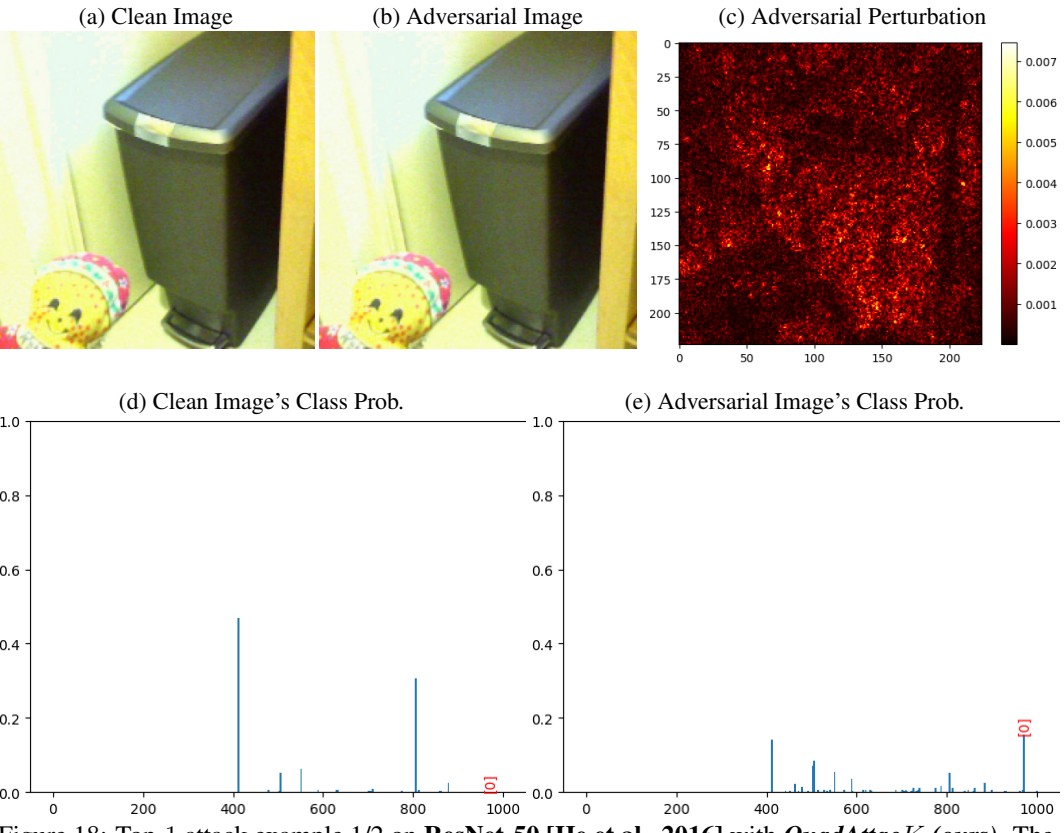

Figure 18: Top-1 attack example 1/2 on **ResNet-50 [He et al., 2016]** with *QuadAttacK (ours)*. The original Top-1 predictions are [ ashcan ]. **The ordered Top-1 targets** (randomly sampled) are: [ eggnog ].

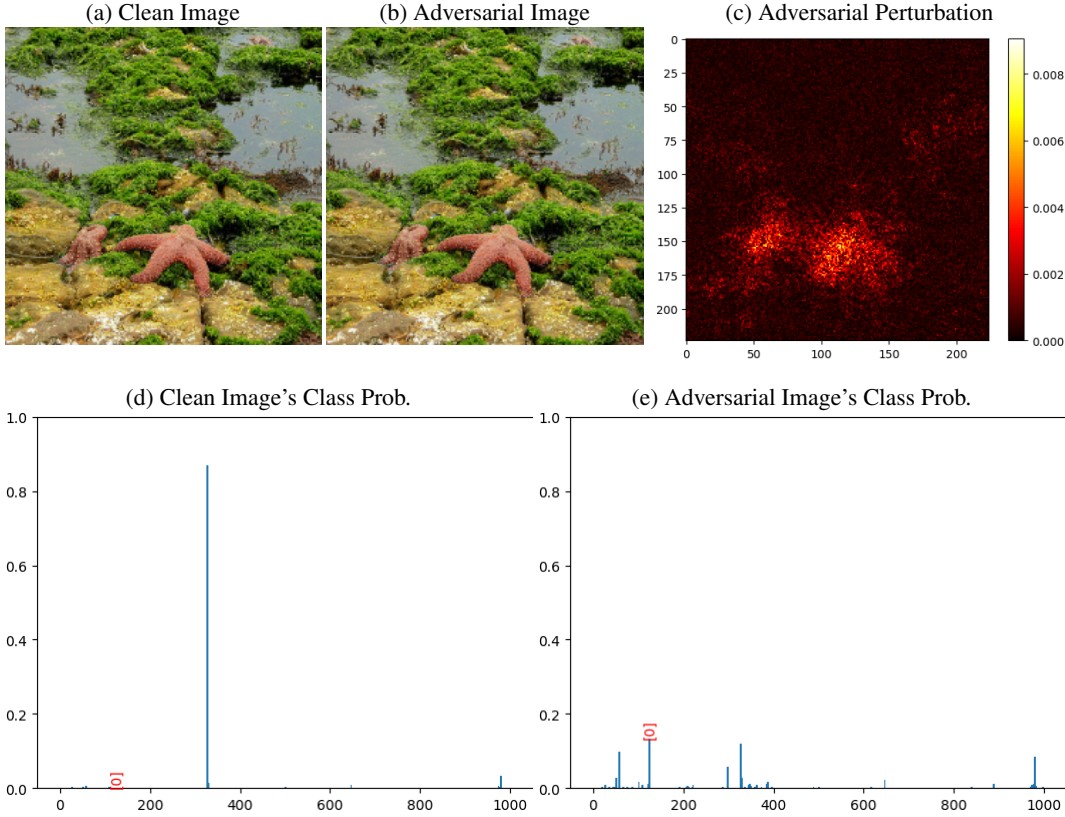

Figure 19: Top-1 attack example 1/2 on **DenseNet-121 [Huang et al., 2017]** with *QuadAttacK (ours)*. The original Top-1 predictions are [ starfish ]. **The ordered Top-1 targets** (randomly sampled) are: [ crayfish ].

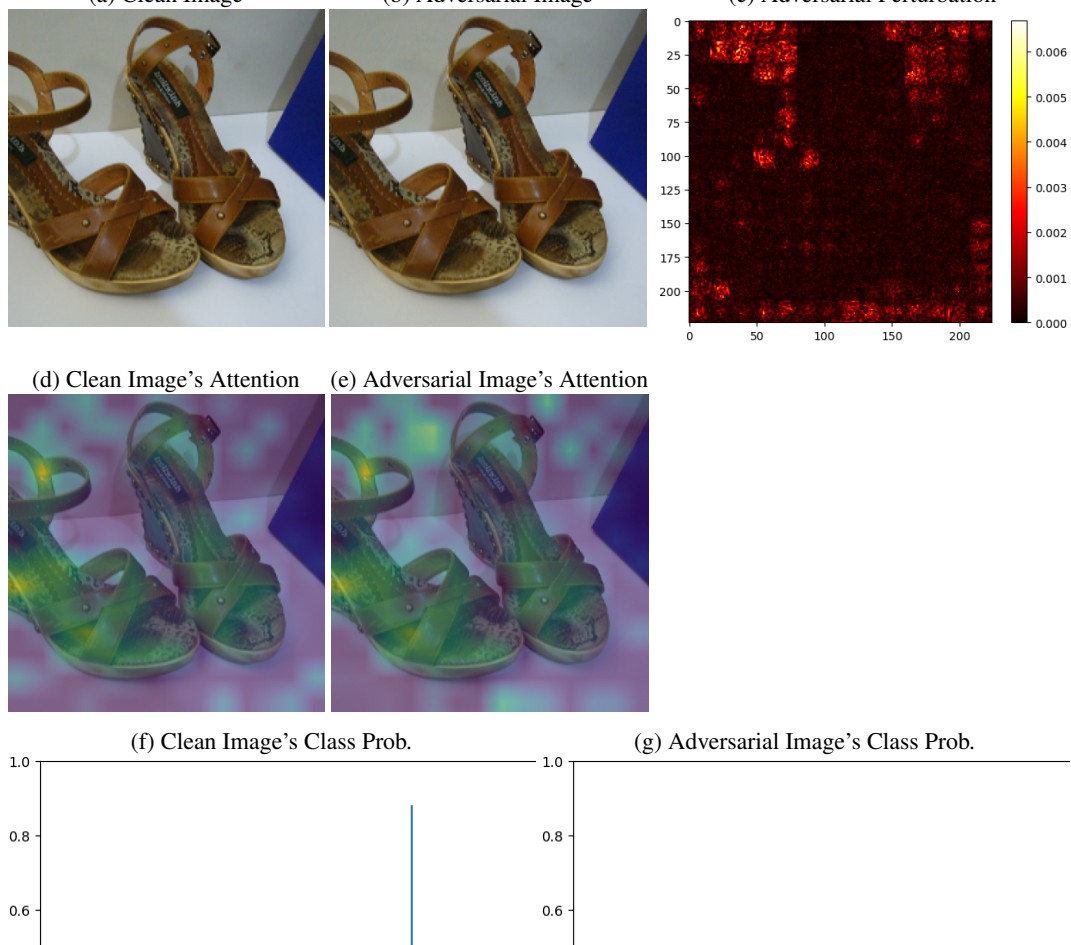

Figure 20: Top-1 attack example 1/2 on **DeiT-S [Touvron et al., 2021]** with *QuadAttacK (ours)*. The original Top-1 predictions are [ sandal    ]. **The ordered Top-1 targets** (randomly sampled) are: [ espresso   ].

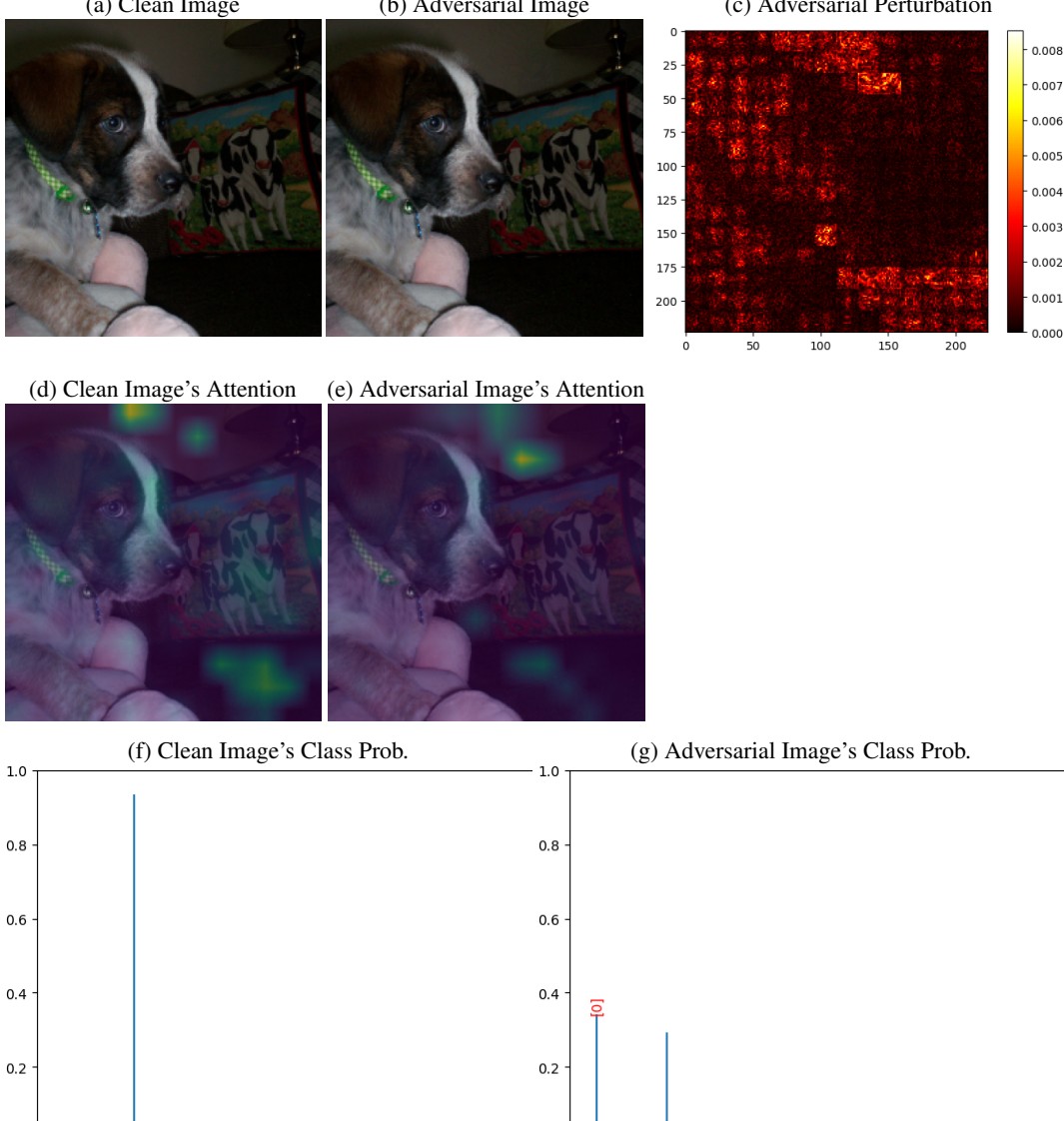

Figure 21: Top-1 attack example 1/2 on **ViT-B [Dosovitskiy et al., 2020]** with *QuadAttacK (ours)*. The original Top-1 predictions are [ `bluetick` ]. **The ordered Top-1 targets** (randomly sampled) are: [ `hen` ].

