# OpenReview forum: "QuadAttac$K$: A Quadratic Programming Approach to Learning Ordered Top-$K$ Adversarial Attacks"
_NeurIPS.cc/2023/Conference — NeurIPS 2023 poster_

### Official Review · Reviewer_wPrk · 2023-06-16

**Soundness:** 3 good
**Presentation:** 3 good
**Contribution:** 3 good
**Rating:** 6
**Confidence:** 3

**Summary:**

The paper proposes a new method for generating top-K adversarial perturbations -- modifying the input such that the classifier predicts the specified K classes, in order. The work addresses this with a two-stage approach, where the first stage computes an adversarial perturbation to the representation, subject to the top-K constraints, and the second stage modifies the input to match this representation.

**Strengths:**

The paper makes a case for top-K adversarial perturbation identification as a step towards more robust systems -- since such perturbations are harder to detect, and a model that is robust to them would be harder to fool. The optimization problem, which is the quadratic objective combined with linear constraints on the non-linear model outputs, is hard to solve hence the proposed two-stage approach which allows the use of methods beyond gradient descent -- specifically, QP on the representation, followed by gradient-based input modification to match this representation.

The method is simple and well-motivated, and comparisons with prior works show that the proposed method is able to perform well where the prior work fails (for the same search budget -- Adversarial Distillation is able to find the adversarial top-K perturbations but at a higher cost).

**Weaknesses:**

- While the paper clearly argues on the value of ordered top-K attacks, it is less clear what might be the alternatives that achieve similar goals but may be easier to optimize

- Second row of (8) seems to have too many non-zeros

- In (12), the last occurrence of D_T is missing a B term



**Questions:**

- What alternatives, if any, are the authors aware of to the ordered top-K objective, which achieves the same goal (attacks that are harder to detect based on the dependence between predicted class probabilities)?

- The paper notes that qpth is a differentiable QP solver. Is the differentiability used?


**Limitations:**

yes

---

> ### Author Rebuttal · Authors · 2023-08-09
>
> Thank you for your time and efforts reviewing our submission. We address your concerns as follows.
>
> **Comment 1:** Alternatives to the Ordered Top-K Attack which are easier to optimize.
>
> > **Response:** As discussed in our global response, ordered Top-$K$ adversarial attacks exploit the principle of \"Class Coherence,\" recognizing the relationships or logic connecting classes within ordinal or nominal frameworks. Unlike unordered or Top-1 attacks, ordered Top-$K$ attacks can subtly manipulate predictions while maintaining coherence within the expected context.
>
> > With respect to alternative easier-to-optimize methods of learning ordered Top-$K$ attacks, one potential direction could be to learn more informative ordered Top-$K$ satisfying distribution to extend the AD method, rather than using the heuristic design method. To that end, a generative model may be trained to become an optimal transport between an easy to sample space with no topological holes onto the space of images that generate class coherent scores (e.g., a bijection from $[-1, 1]^N$ onto the space of images class coherent logit vectors) and a less refined adversarial attack may optimize on the input space of this generative model. By doing this, only perturbations that do not disturb the expected or logical inter-class relationships in the predicted logit vectors would be explored. This would rely on the unlikely requirement that the generative model would span meaningful perturbations, and further a generative model would have to be separately trained for each target model (e.g. ResNet-50).
>
> **Comment 2:** Is the differentiability of qpth used?
>
> > **Response:** Thank you for the very observant question. Yes, we make us of the differentiability of qpth. Imagine the minimizer to Eqn. 6 in our paper is given by the function $\delta_{min} = G(x)$. Then we can phrase our loss as $L = \| x - \hat{x} \|, \hat{x} = x + \delta_{min}$ or equivalently $L = \| \delta_{min} \|$. The differentiability of qpth allows us to directly use the gradient of our perturbation with respect to our feature vector $\frac{d}{dx}[\| \delta_{min} \|]$ for optimization. On the other hand a non-differentiable solver would force
> us to treat $\hat{x}$ as a constant and minimize the loss $L = \| x - C \|$ where $C$ is a constant. While our localized quadratic program is convex, our loss in general is not and thus a loss with a constant target may force use to follow an ever moving target, if we minimize the distance to our solution, then the target may actually become even further in our next iteration. Having access to the gradient $\frac{d}{dx}[\| \delta_{min} \|]$ keeps us from having to rely on a surrogate that may not have a good picture of how the solution the QP itself changes as we move in the search space.
>
> **Comment 3:** Presentation and typos.
>
> > **Response:** Regarding noted issues, our matrix in Eqn. 8 indeed has too many nonzero items in the second row. The item in row 2 column 4 should be a 0 and the whole matrix should be as follows,
> $$\begin{aligned}
>     D_T = \begin{bmatrix}
>     0 & 1 & -1 & 0 & 0; \\
>     -1 & 0 & 1 & 0 & 0; \\
>     1 & 0 & 0 & -1 & 0; \\
>     1 & 0 & 0 & 0 & -1 \\
>     \end{bmatrix},
> \end{aligned}$$
>
> > With respect to a missing term in Eqn. 12, we agree there is a missing bias matrix term. The formulation in line 287 though, is correct. Eqn.12 will be corrected to,
> $$D_T\cdot (A\hat{z} + B) > 0 \quad \Rightarrow \quad -D_T\cdot A\hat{z} \leq D_T\cdot B - \eta$$

---

> > ### Comment · Reviewer_wPrk · 2023-08-14
> >
> > Thank you for the response. I have read it and the rest of the discussion here, and am keeping the original rating.

---

> > > ### Author Response · Authors · 2023-08-14
> > > **Thank you**
> > >
> > > Thank you again for your great efforts and time reviewing our submission and checking our rebuttal.

---

### Official Review · Reviewer_Xdne · 2023-07-05

**Soundness:** 2 fair
**Presentation:** 2 fair
**Contribution:** 3 good
**Rating:** 5
**Confidence:** 4

**Summary:**

It identifies that while sufficient to capture top-K attack constraints, hand-crafted surrogate losses are not necessary and often introduce inconsistency and artifacts in optimization. It eliminates the need of introducing surrogate losses. Instead, it keeps the top-K attack constraints in the vanilla form and cast the optimization problem as quadratic programming (QP). It solves the QP by leveraging a recently proposed differentiable QP layer (for PyTorch).

**Strengths:**

It observes that directly minimizing the lp norm of the learned perturbation together with the hand-crafted surrogate loss could miss the chance of exploiting semantic structures of the feature embedding space ((i.e., the input space to the final linear classifier). Instead, it minimizes the Euclidean distance between the feature embedding vectors at two consecutive iterations in the optimization. This can be understood as the latent perturbation learning versus the raw data perturbation learning. Its proposed latent perturbation learning enables more consistent optimization trajectories in pursuing the satisfaction of the specified top-K attack constraints. The minimized Euclidean distance is then used as the loss together with the lp norm of the learned perturbation in computing the adversarial perturbation via back-propagation at each iteration.

With the proposed QP formulation, it aims to learning top-K attacks efficiently in terms of the computing budget. It eliminates searching the trade-off parameters. Instead, it uses the low-cost 1×S setting for better practicality (e.g. S = 30 or 60), i.e., using a default trade-off parameter. For large K’s (e.g., K > 10), it shows that the QuadAttacK can still achieve appealing ASR, while the prior art completely fails.

**Weaknesses:**

The presentation can be improved. For example, the second row of $D_T$ in equation 8 seems to be incorrect. The expression in equation 12 does not seem to be correct, which is inconsistent with line 287.

It is still not clear how the proposed attack can be applied in practice. Although it lists some advantages of the successful ordered top-K attacks such as some potential directions, it does not provide practical example usages. It is better to discuss the potential practical applications to highlight the importance of the attack.

It mentions that the computation cost of the method is low. However, it does not discuss the complexity or computation cost theoretically. In experiments, all methods seem to adopt 60 steps optimization. It does not really demonstrate that the computation cost is low. Besides, the QP solver may introduce additional costs. The claim to be more efficient with less computation cost may be inaccurate. It is better to provide more discussions or experiments to show the low cost.

For the baselines, it typically needs multiple steps of binary search and a number of iterations of optimization for each trial of binary search (such as 9x30). But in the paper, the baselines do not perform binary search and the configuration is just like 1x30 or 1x60. It is expected that the baselines without binary search does not perform well. Actually the baseline under 9x30 can achieve 100% ASR for top-5 attack. Comparing with these simpler versions of baselines does not demonstrate the method is really better, since the baselines are not used in their original way as designed. Besides, since all baselines and the proposed method use the same 1x30 or 1x60 configuration, and the proposed method has additional cost with the QP solver, it is hard to claim that the proposed method is more efficient with less computations. Maybe it is better to also demonstrate the results under 9x30 for the baselines to show the performance of the full version, and it is also easier to claim efficiency.

**Questions:**

The presentation can be improved. For example, the second row of $D_T$ in equation 8 seems to be incorrect. The expression in equation 12 does not seem to be correct, which is inconsistent with line 287.

It is better to discuss the potential practical applications to highlight the importance of the attack.

For the baselines, it typically needs multiple steps of binary search and a number of iterations of optimization for each trial of binary search (such as 9x30). But in the paper, the baselines do not perform binary search and the configuration is just like 1x30 or 1x60. It is expected that the baselines without binary search does not perform well. Comparing with these simpler versions of baselines does not demonstrate the method is really better, since the baselines are not used in their original way as designed. Besides, since all baselines and the proposed method use the same 1x30 or 1x60 configuration, and the proposed method has additional cost with the QP solver, it is hard to claim that the proposed method is more efficient with less computations. Maybe it is better to also demonstrate the results under 9x30 for the baselines to show the performance of the full version, and it is also easier to claim efficiency.

**Limitations:**

It is better to discuss the potential negative societal impact of this work as it proposes an attack method in deep learning.

---

> ### Author Rebuttal · Authors · 2023-08-09
>
> Thank you for taking your time reviewing our paper. We address your concerns as follows.
>
> **Comment 1:** \"The presentation can be improved.\"
>
> > **Response:** We agree and will carefully revise and proofread the paper.
>
> > Regarding noted issues in our matrix in Eqn. 8, it indeed has too many nonzero items in the second row. The item in row 2 column 4 should be a 0 and the whole matrix should be as follows.
> $$
> \begin{aligned}
>     D_T = \begin{bmatrix}
>     0 & 1 & -1 & 0 & 0; \\
>     -1 & 0 & 1 & 0 & 0; \\
>     1 & 0 & 0 & -1 & 0; \\
>     1 & 0 & 0 & 0 & -1 \\
>     \end{bmatrix}
> \end{aligned}$$
>
> > With respect to a missing term in Eqn. 12, we agree there is a missing bias matrix term. The formulation in line 287 though, is correct. Eqn.12 will be corrected to,
> $$D_T\cdot (A\hat{z} + B) > 0 \quad \Rightarrow \quad -D_T\cdot A\hat{z} \leq D_T\cdot B - \eta,$$
>
> **Comment 2:** \"It is better to discuss the potential practical applications to highlight the importance of the attack.\"
>
> > **Response:** Please refer to the "Elaborated Motivations of Learning Ordered Top-$K$ Adversarial Attacks" in our global response. We will carefully discuss them in the revision.
>
> **Comment 3:** $9\times *$ vs $1\times *$ budgets and efficiency concerns.
>
> > **Response:** While we want to emphasize our claims of efficiency are with respect to adversarial budget (total number of model gradient calls), you raise a great point regarding the need to compare our baseline methods in their original configuration. For this reason, we have added $9\times *$ results for $K=5$ and $K=10$ configurations in Table 1 in our global response PDF. Our QuadAttack$K$ achieves much better results consistently.
>
> > Additionally since $1\times *$ configurations can be tuned to trade ASR for lower energy and vice versa, we compute and plot full ASR vs L2 Energy tradeoff curves for two attack settings.
>
> > With respect the computational overhead of our QuadAttac$K$ method due to solve the QP problem at each iteration. Qualitatively, we do observe that our method is slower than baseline methods. As our response to Comment 3 by the reviewers 2efR and xrmU, we acknowledge the limitation. For a precise understanding of runtime, we have profiled our QuadAttac$K$ and the AD attack on ResNet50 and ViT-B.
>
> >> For ResNet-50 we have found on average QuadAttac$K$ performs 2.47 attack iterations per second whereas AD performs 32.02 iterations per second (a factor of 12.96). For ViT-B QuadAtta$K$ performs 2.96 attack iterations per second whereas AD performs 11.86 iterations per second (a factor of 4).
>
> >> We note that as the target model becomes larger, the adversarial loss constitutes a smaller fraction of total runtime thus the ratio tends toward 1. Further, we note the quicker attack iterations of QuadAttac$K$ on ViT-B which indicates our QP solver converges faster on ViT-B attacks. We will discuss the mean runtimes between different methods in revision.
>
> > To address the overhead of our QuadAttac$K$, we will also explore and compare how the QP solver could be adjusted to initialize the QP solver at the previous iteration's solution to nearly eradicate the cost of the QP solver. To address the overhead of our QuadAttac$K$ , we will also explore and compare how the QP solver could be adjusted to initialize the QP solver at the previous iteration's solution to nearly eradicate the cost of the QP solver.
>
> **Comment 4:** \"It is better to discuss the potential negative societal impact of this work as it proposes an attack method in deep learning.\"
>
> > **Response:** On the one hand, we elaborate in the global response some potential scenarios in practice for which the proposed ordered top-$K$ adversarial attacks may be risky if applied.  On the other hand, since we focus on clear-box attacks, they are less directly applicable in practice compared to opaque-box attacks, which makes the concern less serious. We will make these clear in the Broader Impact section in revision.

---

> > ### Comment · Reviewer_Xdne · 2023-08-14
> > **discussion**
> >
> > Thanks for the comment. My concerns are addressed and I changed my score.

---

> > > ### Author Response · Authors · 2023-08-14
> > > **Thank you**
> > >
> > > Thank you again for your time and great efforts reviewing our submission. We are glad to learn that our rebuttal addressed your concerns.

---

### Official Review · Reviewer_xrmU · 2023-07-06

**Soundness:** 3 good
**Presentation:** 3 good
**Contribution:** 3 good
**Rating:** 5
**Confidence:** 3

**Summary:**

This work proposes a novel approach, QuadAttack to learning ordered top-K adversarial attacks with a low cost. The method is based on a quadratic programming formulation that optimizes the attack objective. Notably, this work extends to a larger K(Top-K). For example, the K is improved from 10 to 15 compared to previous works. QuadAttacK outperforms state-of-the-art methods in terms of attack success rate and query efficiency on various datasets and architectures.

**Strengths:**

1. This work firstly  uses quadratic programming to learn the adversarial attack. The novelty can help this community.
2. The work extends the Top-K attack from Top-10 to Top-15 even larger with a low cost, brining insights to feature works.3
3. When the K=15, the ASR of QuadAttacK is much better than all SOTA methods.
4. Comprehensive results show the generalization of the method.

**Weaknesses:**

1. QuadAttack is not always better than baseline methods. Some deep analysis is lacked.
2. The related work[1] is not mentioned. It could also be considered as a baseline method.
3. It would be better if some results about efficiency are provided.

[1] Zhang, Chaoning et al. “Investigating Top-k White-Box and Transferable Black-box Attack.” 2022 IEEE/CVF Conference on Computer Vision and Pattern Recognition (CVPR) (2022): 15064-15073.

**Questions:**

See above

---

> ### Author Rebuttal · Authors · 2023-08-09
>
> Thank you for your efforts reviewing our submission. We address your concerns as follows.
>
> **Comment 1:** \"QuadAttack is not always better than baseline methods. Some deep analysis is lacked.\"
>
> > **Response:**  Thank you for your detailed review. We appreciate the concern raised regarding the comparisons between QuadAttac$K$ and baseline methods (such as AD), particularly in the context of ASR vs Energy tradeoff.
>
> > We first would like to point out that any mentions of *efficiency* in our paper refer precisely to the adversarial compute budget (total number of model backward passes). With that said, consider the $K=5$ and $1\times 30$ DeiT-S configuration results in our original submission (Table 2). Here, our QuadAttac$K$ obtains an ASR of 0.77 with an L2 energy of 3.34, while AD achieves an ASR of 0.34 with an L2 energy of 2.52. This comparison may give the illusion that while QuadAttac$K$ handles more cases, AD appears as to be a lower-energy attack. To disprove this notion, it's crucial to emphasize that the loss weighting of QuadAttac$K$ can be adjusted (so can the baseline methods'). This loss weight allows trading ASR for lower energy (a trade that is typically exponential). Specifically, if we adjust the weight so that QuadAttac$K$'s ASR is 0.34 in this configuration, its L2 energy will be much lower than that of the AD method. For holistic comparisons in terms of the trade-off between ASR and L2 energy, please refer i) to *Table 1 in our global response PDF* which shows our QuadAttac$K$ consistently outperforms the baseline methods, and ii) to *Figures 1 \& 2 in our global response PDF*, which show that our QuadAttac$K$ significantly outperform the AD method.
>
> > Additionally, we expanded our results to include the $9\times *$ configurations. For every attack instance, this configuration performs 9 binary search steps to determine the lowest possible energy for a successful attack. In other words, this configuration reduces the concept of a tradeoff curve to provide more holistic results since the search eliminates the effect of a loss weight choice. This configuration again shows the large margin between QuadAttac$K$ and its baseline methods.
>
> **Comment 2:** \"The related work [1] is not mentioned. It could also be considered as a baseline method.\"
>
> > **Response:** We really appreciate the additional reference suggestion. We will discuss this excellent work by Zhang, Chaoning et al. in revision. In comparisons, we would like to emphasize that \[1\] is not working on the same problem as our QuadAttac$K$. They investigate a different attack setting as discussed in Section 7 (Discussion) of \[1\]. They refer to our baseline work (AD, Zhang and Wu) and explicitly  state their Top-$K$ optimization definition is not the same problem. To elaborate, we would like to point out there may exist 3 different kinds of \"Top-$K$\" attacks in the literature as follows.
> > -   Untargeted Top-$K$ Adversarial Attack (Easiest): Ground truth shouldn't be in the Top-$K$ classes, Top-$K$ classes can be anything
>     but ground truth.
> > -   Unordered Top-$K$ Adversarial Attack (Harder): Provides specific target Top-$K$ classes that should be in the Top-$K$ predictions
>     after the attack but no particular order of appearance is enforced as long as each target class is somewhere in the Top-$K$
>     predictions.
> > -   Ordered Top-$K$ Adversarial Attack (Hardest): Provides specific target Top-$K$ classes in order and the Top-$K$ predicted classes
>     after attack must match this exact order.
>
> > (Zhang, Chaoning et al.) explore the *Untargeted Top-$K$ Adversarial Attack* whereas we focus on the *Ordered Top-$K$ Adversarial Attack*, so we may not be able to straightforwardly compare with their work.
>
> **Comment 3:** \"It would be better if some results about efficiency are provided.\"
>
> > **Response:** Our QuadAttac$K$ method has a computational overhead to solve the QP problem at each iteration. We acknowledge this limitation in our response to Comment 3 by the reviewer 2efR and in our original submission. For a precise understanding of runtime, we have profiled our QuadAttac$K$ and the AD attack on ResNet50 and ViT-B.
>
> >> For ResNet-50 we have found on average QuadAttac$K$ performs 2.47 attack iterations per second whereas AD performs 32.02 iterations per second (a factor of 12.96). For ViT-B QuadAtta$K$ performs 2.96 attack iterations per second whereas AD performs 11.86 iterations per second (a factor of 4).
>
> >> We note that as the target model becomes larger, the adversarial loss constitutes a smaller fraction of total runtime thus the ratio tends toward 1. Further, we note the quicker attack iterations of QuadAttac$K$ on ViT-B which indicates our QP solver converges faster on ViT-B attacks. We will discuss the mean runtimes between different methods in revision.
>
> > To address the overhead of our QuadAttac$K$, we will also explore and compare how the QP solver could be adjusted to initialize the QP solver at the previous iteration's solution to nearly eradicate the cost of the QP solver.

---

### Official Review · Reviewer_2efR · 2023-07-28

**Soundness:** 3 good
**Presentation:** 2 fair
**Contribution:** 3 good
**Rating:** 5
**Confidence:** 2

**Summary:**

This paper introduces QuadAttackK, a new approach to compute ordered top-K adversarial attacks. The main contribution of this paper is to formulate and efficiently solve the top-K adversarial attack problem via quadratic programming (QP). The experiment results on ImageNet models show that the proposed method improves the attack success rate for large K while maintaining a cheap budget.

**Strengths:**

- Clean formulation + efficient implementation: Introduces a Quadratic Programming (QP) approach to learn ordered top-k clear-box target attacks. The solver leverages recents methods in constrained optimization within neural networks (e.g., based on OptNet) to get an efficient batched QP implmentation.

- State-of-the-art (SOTA) empirical results. The proposed method obtains SOTA attack success rates on ImageNet models such as DenseNet, ResNet and ViTs. It also enables top-k adversarial attacks with large K + low cost budget.


**Weaknesses:**

- The problem of computing *ordered* top-k adversarial attacks lacks some motivation. The motivation in the paper is either too general ("enabling better controllability in learning attacks that are more difficult to defend, revealing deeper vulnerability of a trained DNN") or vague / confusing ("testing the robustness of an attack method itself, especially when K is relatively large").

- Presentation and writing can be significantly improved (e.g., figures and tables are hard to parse, section 4 does not describe the baselines and evaluation metric clearly)

- As noted in the paper, QuadAttackK is slow compared to baselines (need to solve a QP after every iteration) and the attacks do not transfer well to other models.


**Questions:**

None

**Limitations:**

Yes

---

> ### Author Rebuttal · Authors · 2023-08-09
>
> Thank you for your time reviewing our paper. In the following, we address your comments point by point.
>
> **Comment 1:** \"The problem of computing ordered top-k adversarial attacks lacks some motivation.\"
>
> > **Response:** Please refer to the \"Elaborated Motivations of Learning Ordered Top-$K$ Adversarial Attacks\" in our global response. We will carefully discuss them in the revision.
>
> **Comment 2:** \"Presentation and writing can be significantly improved.\"
>
> > **Response:**  Thank you for your recommendations regarding paper presentation. We will carefully revise the paper. Regarding noted issues, our matrix in Eqn. 8 indeed has too many nonzero items in the second row. The item in row 2 column 4 should be a $0$ and the whole matrix should be as follows.
> $$
> D_T = \begin{bmatrix}
>     0 & 1 & -1 & 0 & 0;  \\
>     -1 & 0 & 1 & 0 & 0;  \\
>     1 & 0 & 0 & -1 & 0 ; \\
>     1 & 0 & 0 & 0 & -1  \\
>     \end{bmatrix}
> $$
>
> > With respect to a missing term in Eqn. 12, we agree there is a missing
> bias matrix term. The formulation in line 287 though, is correct. Eqn.
> 12 will be corrected to
> $$D_T\cdot (A\hat{z} + B) > 0 \quad \Rightarrow \quad -D_T\cdot A\hat{z} \leq D_T\cdot B - \eta,$$
>
> > With respect to clearer descriptions of the baselines and evaluation metric, we will revise the paper to make it self-contained regarding those aspects.
>
> **Comment 3:** \"QuadAttac$K$ is slow compared to baselines (need to solve a QP after every iteration) and the attacks do not transfer well to other models.\"
>
> > **Response:**  We acknowledge the limitations raised in terms of the transferability and the QP Solving computational overhead.
>
> > With respect to the transferability of learned attacks,  we deliberately chose to focus on the complexity of this optimization problem in the clear-box setting, and on the learnability of ordered top-$K$ attacks, especially for large $K$'s. We notice that learning transferrable clear-box attacks is a challenging problem even for the traditional top-$1$ setting. We leave the study of attack transferability as a future endeavor. One potential starting point could be to investigate the problem of how to apply our QuadAttac$K$ for multiple different networks simultaneously. Another potential direction is to first gain better understanding of the alignment of the latent spaces (the input space to the linear classifier) between different networks, and then to guide the learning of attacks to focus more on those aligned sub spaces.
>
> >With respect to the overhead of solving a QP problem at every iteration, our QuadAttac$K$ method has a computational overhead to solve the QP problem at each iteration. We acknowledge this limitation in our response to other reviewers and in our original submission. For a precise understanding of runtime, we have profiled our QuadAttac$K$ and the AD attack on ResNet50 and ViT-B.
>
> >> For ResNet-50 we have found on average QuadAttac$K$ performs 2.47 attack iterations per second whereas AD performs 32.02 iterations per second (a factor of 12.96). For ViT-B QuadAtta$K$ performs 2.96 attack iterations per second whereas AD performs 11.86 iterations per second (a factor of 4).
>
> >> We note that as the target model becomes larger, the adversarial loss constitutes a smaller fraction of total runtime thus the ratio tends toward 1. Further, we note the quicker attack iterations of QuadAttac$K$ on ViT-B which indicates our QP solver converges faster on ViT-B attacks. We will discuss the mean runtimes between different methods in revision.
>
> > To address the overhead of our QuadAttac$K$, we will also explore and compare how the QP solver could be adjusted to initialize the QP solver at the previous iteration's solution to nearly eradicate the cost of the QP solver.

---

> > ### Comment · Reviewer_2efR · 2023-08-14
> > **Update**
> >
> > Thanks for the response. I read the rebuttal and I would like to keep my score as is.

---

### Author Rebuttal · Authors · 2023-08-09

We sincerely thank all reviewers for their constructive feedbacks which help us to greatly improve our submission.  We first address some common concerns.

**Drastically Improved Results.** Please refer to *Table 1 in our global response PDF* for the improved results. In optimization, perturbations are initialized with some small $\epsilon$ energy white gaussian noise. During the initial steps of optimization, the optimizer takes steps with large increases in perturbation energy since $\epsilon$ happens to be away from many required energies for a successful attack. These large increases in energy induces a momentum in the $AdamW$ optimizer, which makes it difficult to reduce L2 energy in future iterations even if our objective function's gradient points towards a direction with minimal energy. By introducing a small number of warmup steps (e.g., 5, as commonly done in training a network on ImageNet from scratch) after which the optimizer's state is reset, we have managed to improve the performance of all analyzed methods. Our QuadAttac$K$ benefits most.


**Analyses on the Trade-Off Between Attack Success Rates and Attack Energies.** Please refer to *Figures 1 \& 2 in our global response PDF*. These tradeoff curves explore the concept of how a higher success rate may be achieved by choosing to have higher energies and conversely a lower energy may be achieved by choosing to have a lower success rate. Nonetheless, these curves holistically compare the capacity of QuadAttac$K$ against the baseline method.
> Additionally, we expanded our results to include the $9\times \*$ configurations. For every attack instance, this configuration performs 9 binary search steps to determine the lowest possible energy for a successful attack. In other words, this configuration reduces the concept of a tradeoff curve to provide more holistic results since the search minimizes the effect of a loss weight choice. This configuration again shows the large margin between QuadAttac$K$ and its baseline methods.

**Elaborated Motivations of Learning Ordered Top-$K$ Adversarial Attacks:**
These attacks exploit the principle of \"Class Coherence\", recognizing the relationships or logic connecting classes within ordinal or nominal frameworks. Unlike unordered or Top-1 attacks, ordered Top-$K$ attacks can subtly manipulate predictions while maintaining coherence within the expected context.

In an ordinal context like credit ratings (*\[Extremely High Risk, Very High Risk, High Risk, Moderate Risk, Low Risk, Minimal Risk, No
Risk\]*), they can downgrade a rating without disrupting the logical flow, making it less detectable. In a nominal context such as a
recommendation system attacking predicted user interests in shopping classes from *\[Books, Movies, Beauty, Furniture, Fashion\]* into
*\[Fashion, Books, Movies, Beauty, Furniture\]* in order; subtly pushing Fashion onto the user while keeping them engaged with their true
interests.

An unordered change might disrupt these logical groupings, (e.g. push Fashion to the top, but also inadvertently pushing Furniture up as well) leading to a clumsy attack, raising a red flag to a human user, or even trivial detection in a system that validates class coherence. These refined and subtle manipulations can have diverse consequences, from affecting financial decisions to subtly influencing user behavior in online platforms. They are more challenging to detect than other types of attacks, both quantitatively or from a human standpoint, and represent a nuanced and significant threat that warrants deeper exploration in adversarial machine learning. For completeness, we provide more specific examples below.

> Ordinal example: Imagine a cancer risk assessment tool that analyzes 2D medical images like mammograms to categorize patients' cancer risk into the ordinal 7-level risk ratings (as the credit ratings). An oncologist could use this tool to triage patients, prioritizing those in the highest risk categories for immediate intervention. An attacker aiming to delay treatment might use an ordered top-5 adversarial attack to change a prediction for a patient initially assessed as Very High Risk. They could target the classes *\[High Risk, Moderate Risk, Low Risk, Minimal Risk, Very High Risk\]*, subtly downgrading the urgency without breaking the logical sequence of risk categories. An unordered attack, in contrast, might lead to a sequence like *\[Low Risk, Very High Risk, Minimal Risk, Moderate Risk, High Risk\]*, disrupting the ordinal relationship between classes. Such a disruption could raise red flags, making the attack easier to detect.

> Nominal class example: Traffic control systems could use deep learning to optimize flow by adjusting the timing of traffic lights based on the types of vehicles seen. Priority might be given to certain vehicle classes, such as public transit or emergency vehicles, to improve
response times. Imagine a city's traffic control system, which has specific traffic light timing behavior for the nominal vehicle
categories *\[Emergency Vehicle, Public Transit, Commercial Vehicle, Personal Car, Bicycle\]*. Public transit might be given slightly
extended green lights during rush hours to encourage public transportation use. An attacker wanting to cause delays for personal
cars without raising alarms could launch an ordered Top-2 adversarial attack, targeting the sequence *\[Commercial Vehicle, Public
Transit\]*. This would cause the system to interpret most personal cars as commercial vehicles during the attack, applying the extended green light times meant for public transit to lanes primarily used by commercial vehicles. An unordered top-2 attack that may result in
*\[Emergency Vehicle, Commercial Vehicle\]*, would likely be quickly detected, as emergency vehicle priority changes are significant and
could be easily noticed by traffic operators (this weakness is exacerbated in any Top-1 attack or unordered attacks).

---

### Decision · Program_Chairs · 2023-09-21

**Decision:**

Accept (poster)

**Comment:**

This paper proposes a new method for constructing top-k adversarial examples. Reviewers found the the method interesting, and the results compelling. The authors constructively engaged with the reviewers during the rebuttal period. There are some justified concerns about the quality of presentation and writing. Authors, make sure to implement the promised changes to address reviewer concerns.